# β-Lactam Antibiotics and β-Lactamase Enzymes Inhibitors, Part 2: Our Limited Resources

**DOI:** 10.3390/ph15040476

**Published:** 2022-04-13

**Authors:** Silvana Alfei, Anna Maria Schito

**Affiliations:** 1Department of Pharmacy (DIFAR), University of Genoa, Viale Cembrano, 4, 16148 Genoa, Italy; 2Department of Surgical Sciences and Integrated Diagnostics (DISC), University of Genoa, Viale Benedetto XV, 6, 16132 Genoa, Italy; amschito@unige.it

**Keywords:** β-lactam antibiotics (BLAs), β-lactamase enzymes (BLEs), serine β-lactamase enzymes (SBLEs), metallo β-lactamase enzymes (MBLEs), Ambler classification, carbapenemases, β-lactamase enzymes inhibitors (BLEsIs), clinically approved BLEsIs, clinical and preclinical trials

## Abstract

β-lactam antibiotics (BLAs) are crucial molecules among antibacterial drugs, but the increasing emergence of resistance to them, developed by bacteria producing β-lactamase enzymes (BLEs), is becoming one of the major warnings to the global public health. Since only a small number of novel antibiotics are in development, a current clinical approach to limit this phenomenon consists of administering proper combinations of β-lactam antibiotics (BLAs) and β-lactamase inhibitors (BLEsIs). Unfortunately, while few clinically approved BLEsIs are capable of inhibiting most class-A and -C serine β-lactamases (SBLEs) and some carbapenemases of class D, they are unable to inhibit most part of the carbapenem hydrolyzing enzymes of class D and the worrying metallo-β-lactamases (MBLEs) of class B. Particularly, MBLEs are a set of enzymes that catalyzes the hydrolysis of a broad range of BLAs by a zinc-mediated mechanism, and currently no clinically available molecule capable of inhibiting MBLEs exists. Additionally, new types of alarming “superbugs”, were found to produce the New Delhi metallo-β-lactamases (NDMs) encoded by increasing variants of a plasmid-mediated gene capable of rapidly spreading among bacteria of the same species and even among different species. Particularly, NDM-1 possesses a flexible hydrolysis mechanism that inactivates all BLAs, except for aztreonam. The present review provides first an overview of existing BLAs and the most clinically relevant BLEs detected so far. Then, the BLEsIs and their most common associations with BLAs already clinically applied and those still in development are reviewed.

## 1. Introduction

The β-lactam antibiotics (BLAs) are the most common antibacterial molecules recommended and used for counteracting a large variety of infectious diseases [1]. BLAs are characterized by a bicyclic or monocyclic structure both comprising a four-term β-lactam nucleus, which represents the weak point of these drugs. Indeed, the β-lactam amide group can undergo enzymatic hydrolysis, under the action of β-lactamases enzymes (BLEs) thus providing compounds free of antibacterial activity [2]. Taking natural penicillin as a template molecule, several penicillin derivatives, which now form different classes of β-lactam compounds, have been developed in the last century to improve potency, spectrum of activity, pharmacokinetic and safety profiles and to respond to the rise of bacterial resistance [1]. Currently, the main classes of β-lactams in clinical use include molecules characterized by a bicyclic nucleus, such as penicillin-like BLAs, cephalosporins, and carbapenems, or monocyclic systems as the monobactams. Notably, in the group of narrow and broad-spectrum penicillin-like BLAs, which represent the 39.7% of prescriptions in the United States (USA), the four-membered β-lactam ring is fused to a thiazolidine ring [1,2,3] (Figure 1a). In cephalosporins (47.5% of prescriptions in the USA), a six-membered dihydrothiazine ring is fused with the β-lactam one (Figure 1b), while in carbapenems (11.2% of prescriptions in the USA), the bicyclic system is completed by a five-membered pyrroline (Figure 1c) [1,2,3]. Differently, in the monocyclic monobactams, representing only 1.66% of the prescriptions in the USA, the β-lactam nucleus possesses a sulfonic acid group on the nitrogen atom of the cyclic amide (Figure 1d) [3].

The pioneer molecules of each class originally were natural products (penicillin in 1929, cephalosporins by Newton and Abraham in 1954, olivanic acid alias carbapenem by Brown and co-workers in 1976, and monobactams by Sykes and Imada and their respective co-workers in 1981), which, upon extensive modifications, provided an array of semi-synthetic derivatives [3,4].

Figure 2 shows a different classification of the bicyclic core structures of β-lactams based on the presence of 2,3 double bonds (C=C) in the five or six terms ring fused on the β-lactam one and on the type of atom in positions 4 and 5 on the five-termed or six-termed ring, respectively. Particularly, three families, namely penams (a), penems (b), cepham (c), and cephem (d) include, penams (1), carbapenams (2), and oxapenams (3), penems (4), carbapenem (5), and oxapenem (6), cephams (7), carbacepham (8), and oxacepham (9), and finally, cephems (10), carbacephem (11), and oxacephem (12) [4]. 

Generally, β-lactam antibiotics (BLAs), while relatively safe, have a broad spectrum of antibacterial potency, being active both on clinically relevant Gram-positive and Gram-negative species [4]. Their activity is based on the capability to inhibit a set of transpeptidase enzymes, known as penicillin-binding proteins (PBPs), that are essential for the synthesis of the peptidoglycan layer of the bacterial cell wall [5]. In the absence of peptidoglycan, bacteria cannot grow and consequently die [6]. 

While penicillin-like BLAs and cephalosporins display a wide range of activity against both Gram-positive and Gram-negative bacteria, the monobactams, deprived of the fused rings system, are active specifically against aerobic Gram-negative pathogens [7,8]. Differently, carbapenems possess a potent, broad spectrum of activity against both Gram-positive and Gram-negative bacteria, and work on strains against which the most part of other antibiotics are no longer functioning, thus representing an essential group of antibiotics for the treatment of infections caused by multidrug-resistant (MDR) bacteria [9]. Unfortunately, to resist the action of BLAs, often inappropriately used in different clinical settings, they are increasingly developing different defense mechanisms, which now represent a serious threat [10,11].

The mechanisms by which bacteria develop resistance to BLAs are diverse and include hyper-expression of efflux pumps, reduced permeability due to reduced expression of porins, production of altered PBPs, and BLA inactivation by delivering BLEs. 

As an example, the alteration of the sequence of transpeptidases, target of BLAs, by mutation or recombination, thus creating enzymes with low affinity for BLAs, is an important source of resistance in *Streptococcus pneumoniae* [10]. Additionally, the acquisition of new transpeptidases, such as in the case of PBP2a, that bind β-lactam antibiotics gently is the foundation for developing resistance in methicillin-resistant *Staphylococcus aureus* (MRSA) or in other species of the same genus [10]. However, despite these important examples, the production of BLEs is the most common mechanism of bacterial resistance to BLAs [4].

Particularly, BLEs are bacterial enzymes that hydrolyze the β-lactam amide bond present in the four-term cycle of BLAs, thus causing the ring opening and the generation of molecules to no longer be effective [6]. BLEs are the highest cause of resistance, particularly, among pathogens of Gram-negative species, but also certain Gram-positive species are alarming producers of BLEs According to the Ambler classification, the primary sequence homology, and the differences in hydrolytic mechanisms, BLEs are divided into four classes (classes A, B, C, and D) belonging to two families [4,12,13,14]. 

Particularly, BLEs of classes A, C, and D form the family of the active-site serine BLEs (SBLEs) [15], while class-B enzymes constitute that of metallo-BLEs (MBLEs), which are the enzymes that are the main worry when it comes to world health, as they possess a broad spectrum of actions against which no remedy is currently clinically approved and used in therapy works [16,17]. Moreover, SBLEs, while initially active only on penicillin-like BLAs, following progressive mutations, have emerged as extended-spectrum serine BLEs (ESSBLEs) capable of also hydrolyzing the β-lactam ring of cephalosporins [4]. 

To address this condition, the clinical use of carbapenems, which possess higher stability towards BLEs in addition to having a wider spectrum of action, strongly increased in the 2000s [18,19,20,21]. Initially, carbapenems were resistant to the inactivation carried out by SBLEs and ESSBLEs, acting even as inhibitors for many of these enzymes [20]. Unfortunately, the continued and excessive use of BLAs, which is frequently inappropriate, after having triggered the spread of resistance to first-generation antibiotics and then to the extended-spectrum cephalosporins (cefotaxime, ceftriaxone, and ceftazidime) in the past decade, has selected bacteria producing carbapenemases capable of also hydrolyzing carbapenems. Such bacteria are resistant to the most clinically relevant carbapenems, including doripenem, ertapenem, imipenem, meropenem, biapenem, panipenem, razupenem, and tomopenem [9,22,23,24,25,26]. Although the bacterial resistance to carbapenems mainly depends on the carbapenemase activity strictly connected to the hyperexpression of AmpC, other elements—such as the augmented presence of efflux pumps, the diminished number of porins, or/and their diameter, and mutations in PBPs (PBP2 or PBP3)—are also involved in the resistance to such antibiotics [24,27,28]. 

Specifically, the most worldwide health public concern is raised by the proliferation of carbapenem-resistant bacteria, which produce *Klebsiella pneumoniae* carbapenemases (KPCs) belonging to the ESSBLEs of class A, New Delhi MBLEs (NDMs) belonging to class B, and oxacillinases (OXAs) belonging to the ESSBLEs of class D [4,14].

The World Health Organization (WHO), in the year 2017, published a list of the most alarming pathogens for which available treatments are no longer effective. Among the bacteria included in the list, carbapenem-resistant *Acinetobacter baumannii*, *Pseudomonas aeruginosa*, and carbapenem- and third-generation cephalosporin-resistant *Enterobacteriaceae*, were inserted in the highest category of urgency [29]. The global effort to prevent resistance to BLAs is focused on the development of broad-spectrum β-lactamase enzyme inhibitors (BLEsIs). Except for those recently developed and approved, which do not structurally resemble BLAs, the available clinically applied BLEsIs have the β-lactam core, which is recognized by BLEs. β-lactam BLEsIs bind to BLEs in a suicidal and irreversible way, thus impeding them from interacting with BLAs and inactivating them [4]. BLEs producing bacteria are currently treated with BLEsIs administered in proper combinations with BLAs [23]. Surely, the discovery and subsequent commercialization of clavulanic acid, sulbactam, and tazobactam and their clinical application in combinations such as amoxicillin/clavulanic acid, ampicillin/sulbactam, and piperacillin/tazobactam has strongly improved the alarming scenario of the antimicrobial therapy, allowing the recovery of BLAs otherwise no longer functional [4]. Although they have been used over a long period, new approaches are needed and are in progress to develop and expand the number of BLEsIs and their spectrum of action [4,14,23]. Their limited spectrum of inhibition involving mainly the BLEs of class A, including also ESSBLEs but not including carbapenemases; their slight inhibition of BLEs of class C, as well as the increasing emergence and propagation of class B and class D carbapenemases; make an incessant search for new inhibitors necessary. Recently, new groups of BLEsIs such as the diazobicyclooctanes (DBOs) and boronic acid derivatives have emerged [30]. Notably, avibactam and relebactam among DBOs and vaborbactam among the boronic acid derivatives have been approved by the FDA and are now clinically applied [12,31]. Unfortunately, despite the development and commercialization of these new inhibitors, new compounds able to inactivate carbapenemases, especially those of class B, are still required to improve the limited, and sometimes the lack of, therapeutic options to counteract carbapenemase-producing bacteria. The present review provides, first, an overview of the existing BLAs and the most clinically relevant BLEs detected so far. Then, it provides an “in a round” scenario concerning the main BLEsIs developed till now, their most common combinations with BLAs, and their current or potential clinical use, also giving information concerning the route of administration and pharmacokinetics. Particular attention has been paid to the carbapenemases inhibitors recently approved for clinical use, at clinical or preclinical stages of development, and to those still at the very early stages of experimentation. Particularly, we considered those compounds that exhibited high activity against the most widely distributed and the most clinically relevant BLEs of classes A, B, C and D. Finally, to summarize the state of the art concerning BLEsIs herein reviewed, we originally organized the conclusion section, providing graphical representations of the most important information reported in this paper, thus giving the reader a scenario graspable at a glance. As the first step to fully creep into the topic of this review, Table 1 gives an idea of the current armamentarium available for the treatment of bacterial infections [4].

## 2. β-Lactamase Enzymes (BLEs)

The production of BLEs is the most common strategy implemented both by Gram-positive and Gram-negative bacteria to develop resistance to BLAs [4]. Bacteria that can produce BLEs include but are not limited to *M. tuberculosis*, *S. aureus*, MRSA, *Enterobacteriaceae* such as *K. pneumoniae*, *Citrobacter* spp., *Proteus vulgaris*, *Morganella* spp., *Salmonella* spp., *Shigella* spp., *E. coli*, *H. influenzae*, *N. gonorrhoeae*, and non-fermenting Gram-negative bacteria including *P. aeruginosa* and *A. baumannii* [4,23]. Several Gram-negative bacteria naturally produce BLEs encoded by chromosomal genes, whose original role was to defend microorganisms from other bacteria capable of producing BLAs, as well as to remove molecules structurally, such as the BLAs that act as regulators of cell wall synthesis [1].

According to the primary sequence homology and the Ambler classification, BLEs have been arranged into four classes, namely A, B, C, and D [32]. BLEs of classes A, C, and D belong to the SBLEs, which catalyze the hydrolysis of BLAs, via a serine-bound acyl ester intermediate [4,15]. The mechanism of action of SBLEs has been reported in Figure 1. Particularly, the acyl ester is formed thanks to a nucleophilic acyl substitution, promoted by the attachment of the hydroxyl group of the serine present in the active site of the enzyme to the carbonyl group of the β-lactam ring of BLAs, thus causing its opening (Figure 1). Next, the nucleophilic attack by a water molecule cleaves the covalent bond between the enzyme and the carbonyl group of the β-lactam ring generating a carboxylic group, thus allowing the degraded BLA to spread elsewhere, allowing the newly free enzyme to inactivate new β-lactam molecules [4]. Also, a decarboxylation can occur, thus leading to the complete and irreversible inactivation of BLAs [4] (Figure 1). 

Differently, the class-B enzymes belonging to the MBLEs cleave the β-lactam ring of BLAs by a mechanism like that of metalloproteases, which are protease enzymes whose catalytic mechanism involves a metal [4]. 

The metal ion involved in the mechanism of the catalytic activity of MBLEs is zinc (Zn^2+^), which is coordinated with the protein through three ligands, variable between histidine, glutamate, aspartate, lysine, and arginine. The fourth coordination position is occupied by a labile hydrolyzing hydroxyl group, responsible for hydrolysis, which is activated by the bivalent cation [4]. The MBLEs are grouped into subclasses B1, B2, and B3, which are differentiated by their Zn^2+^ coordinating ligands and the number of metals present in the active site. Generally, two Zn^2+^ (Zn1 and Zn2) are in the site of active enzymes of subclasses B1 and B3, and one in that of enzymes of subclass B2. The chemical mechanism by which the di-Zn MBLEs of subclasses B1 and B3 hydrolyze carbapenems is shown in Figure 2 [33].

Briefly, Zn1 and Zn2 coordinated with the hydroxyl ion, first lose coordination with two of the three ligands that anchor the active site to the protein and interact with the carbapenem by coordinating its carbonyl and its exocyclic carboxylate, thus both polarizing the β-lactam carbonyl and anchoring the antibiotic in the active site. Secondly, the carbonyl is attacked by the OH ion forming a tetrahedral adduct [33]. This unstable intermediate subsides to open the ring, generating a nitrogen anion, which is stabilized by one of the Zn^2+^. Protonation of the anion followed by the release of the product and rebinding of a water molecule completes the catalytic cycle (Figure 2) [33]. 

Treatment with chelating agents, such as EDTA, leads to complete inactivation of the enzyme, due to its chelating capability that removes zinc.

During the hydrolytic process by MBLEs, no covalent intermediate is formed, and this is the reason for the ineffectiveness of the currently clinically approved BLEsIs, which work by binding BLEs covalently. Regarding this, the spread of bacterial strains expressing MBLEs, such as the NDM-1, has engendered considerable concerns [34]. 

Note that, SBLEs belong to a larger family of penicillin-recognizing enzymes (PREs) that also include the transpeptidases that crosslink bacterial cell walls (PBPs). All of these enzymes encompass the serine active site and a conserved triplet of amino acids, interposed between the serine active site and the C terminus [35]. The three-dimensional (3D) crystal structures of several class-A, -C, and -D BLEs and those of transpeptidases are similar, thus suggesting a common evolutionary origin for PREs [4] On the contrary, the 3D structures of several class-B enzymes lack of similarity with the SBLEs and transpeptidases, thus indicating a different and independent evolutionary origin [4]. 

Globally, more than 400 different types of BLEs have been reported. Table 2 collects some of the most recognized and clinically relevant BLEs and the superbugs that typically produce them.

TEMs such as TEM-2, TEM-3, and TEM-4 are the most common SBLEs of class A whose genes are present in plasmids of Gram-negative bacteria such *Enterobacteriaceae*, *P. aeruginosa*, *H. influenzae*, and *N. gonorrhoeae*. Due to punctiform mutations in single amino acids, currently, 140 BLEs of TEM type have been reported [37]. These enzymes are capable of hydrolyzing penicillin’s β-lactam ring and that of cephalosporins belonging to the first and second generations but are ineffective on cephalosporins of the third generation, such as cefotaxime, ceftazidime, and ceftriaxone, and against those of the fourth generation, such as cefepime. 

The ESBLEs of class A named SHVs of which 60 varieties are known, although currently found worldwide, are predominant in Europe and the United States [38,39,40,41]. They are produced by several members of *Enterobacteriaceae* and by *P. aeruginosa* and are capable of hydrolyzing aztreonam and third-generation cephalosporins, including cefotaxime and ceftazidime. On the contrary, SHVs are inhibited by clavulanic acid, and bacteria producing SHVs are susceptible to cefoxitin and imipenem. 

Differently, cefotaxime and other oxyimino β-lactams substrates, including ceftazidime, ceftriaxone, and cefepime are hydrolyzed by the enzymes belonging to the CTX–M family. There are more than 40 class-A ESSBLEs of CTX-M type, which are commonly produced by isolates of *Salmonella enterica serovar*
*typhimurium* and of *E. coli*, but are also engendered by some species of *Enterobacteriaceae*. CTX-Ms including CTX- M-14, CTX-M-3, and CTX-M-2, CTX-Ms are the main ESSBLEs in South America, which prevailed over TEM and SHV during the first decade of the 21st century [38,39,40,41].

Class-A carbapenemases of the KPC family are coded in plasmids, are the most widely distributed worldwide, and are predominantly found in *K. pneumoniae*. However, KPCs have also been identified in *Enterobacter* spp., *Salmonella* spp., *P. aeruginosa*, and *A. baumannii* [36]. Curiously, concerning KPCs, two strains of KPC-producers *K. pneumoniae*, resistant to the combination ceftazidime–avibactam (CZA), were recently isolated from infected patients never treated with CZA [52]. 

The whole-genome sequencing characterization showed that both isolates had altered outer membrane porins and novel pKpQIL plasmid derivatives (pIT-14159 and pIT-8788), harboring two copies of the KPC-3-encoding transposon. Plasmid pIT-8788 was a cointegrate of pIT-14159, which apparently evolved in vivo during infection. When transferred to *E. coli* DH10B, originally susceptible to CZA, pIT-8788 increased the MIC value of CZA 32-fold [52].

Several *Enterobacteriaceae* and a few other bacteria expressing the chromosomal AmpC gene produce the class-C AmpC β-lactamases that are capable of inactivating cephalosporins, including cephalothin, cefazolin, cefoxitin, several penicillin-like BLAs, and many combination BLEsIs/BLAs. The inducted overexpression of the AmpC gene can provide many microorganisms with resistance to broad-spectrum cephalosporins, including cefotaxime, ceftazidime, and ceftriaxone. Additionally, in *E. coli*, *K. pneumoniae*, and *P. mirabilis*, which are missing the chromosomal AmpC gene or poorly express it, transmissible plasmid encoding genes for AmpC enzymes can appear [44]. 

Antibiotics of the carbapenem family are usually effective for treating infections caused by bacteria producing AmpCs, but resistance to carbapenems has been observed in some organisms, due to other phenomena associated with the overexpression of the AmpC gene, such as reduction of outer membrane porins or enhancement of the antibiotics’ efflux by activation of efflux pumps [38,39,40,41].

Generally, AmPCs do not provide resistance to carbapenems, but five enzymes belonging to this group recently identified and named ACT-1, DHA-1, CMY-2, CMY-10, and ADC-68, are capable of hydrolyzing carbapenems [45].

The OXA-type ESSBLEs belonging to class D, except for OXA-18, are carried on plasmids. Bacteria producing OXAs are resistant to ampicillin and cephalothin and are capable of hydrolyzing oxacillin, cloxacillin, and carbapenems. Such resistant bacteria, except for *Enterobacteriaceae*, which produce OXA-48 and OXA-48-like enzymes, become susceptible to the above-mentioned antibiotics if administered in combination with clavulanic acid. Bacteria producing OXA-17 possess high resistance to cefotaxime and cefepime [38,39,40,41]. Interestingly, although the discovery of OXA-type carbapenemases (also named carbapenem hydrolyzing class-D lactamase (CHDLs)) dates to many years ago, their rapid spread is recent [42,43].

Although OXA-type ESSBLEs such as OXA-51, OXA-23, and OXA-58 have been found mainly in *P. aeruginosa*, they are prevalent also in *Acinetobacter* spp. Particularly, OXA-23-like, OXA-58-like OXA-24/40-like, OXA-143-like, and OXA-235-like ESSBLEs are responsible for the emergence of resistance to carbapenems in *A. baumannii* [23].

Other enzymes such as *Pseudomonas* extended resistance (PERs); Vietnam extended-spectrum (VEBs); *Serratia fonticola* (SFO-1); Tlahuicas (Indian tribe) (TLA-1); Brazilian ESSBLEs (BES-1); GESs; Belgium ESSBLEs (BEL-1); and TLA-2, which has a 51% amino-acid identity with TLA-1 are plasmid-mediated ESSBLEs produced mainly in *P. aeruginosa* and less common than others [38,39,40,41].

Prior to the mid-1990s, the MBLEs of class B were found only in environmental, poorly pathogenic bacterial species. Such enzymes were encoded by a non-transferable chromosomal gene. Subsequently, several MBLEs encoded by plasmids, which facilitate their transfer among bacteria and are capable of hydrolyzing and inactivating carbapenems, were instead found in strains of Gram-negative bacteria including *P. aeruginosa*, *A. baumannii* and *Enterobacteriaceae* [46]. According to similarities in their sequences, MBLEs are grouped into subclasses B1, B2, and B3 [33]. While the enzymes of subclass B2 efficiently catalyze the hydrolysis of carbapenem antibiotics but possess low activity against penicillin and cephalosporins, those of subclasses B1 and B3 also hydrolyze penicillin-like BLAs and cephalosporins. Among MBLEs, IMPs, VIMs, GIMs, SIMs, SPMs, and NDMs represent the most common families [33,46]. The first transferable MBLE was IMP-1 (imipenem hydrolyzing enzyme), which was found in an isolate of the *P. aeruginosa* species in Japan in 1988. Subsequently, IMPs were reported in Europe (1997), Canada, and Brazil, and presently they can be detected worldwide, beyond *P. aeruginosa* also in *Enterobacteriaceae* [46]. VIM-1 (Verona integron-encoded metallo-β-lactamase-1), which includes 14 varieties of enzymes now globally distributed, was found in Verona, Italy, in 1997. While rare in *Enterobacteriaceae*, VIM-type MBLEs are commonly produced by *P. aeruginosa* and *P. putida* [46].

SPM-1 (Sao Paulo MBLE), GIM-1 (German imipenemase), and SIM-1 (Seoul imipenemase) were isolated in Sao Paulo (Brazil) in 1997, in Germany (2002) and in Korea (2003), respectively [46].

Worryingly, bacteria carrying the New Delhi metallo-β-lactamase-1 (NDM-1) resistance gene, first isolated in India and now spread worldwide, are considered as new types of “superbugs” capable of inactivating almost all BLAs, except for aztreonam [47,48].

Particularly, NDM-1 belongs to the B1 subclass of MBLEs of group B and is composed of a single polypeptide chain of 27.5 KDa. Three loop regions—L3, L7, and L10—form the main part of the active site, which allocates the two zinc ions (Zn1/Zn2) [49,50,51]. It has been recognized that the broad-spectrum activity of NDM-1 is due to the presence of a wide and shallow substrate-binding capsule cavity that is capable of easily accepting different types of BLAs. NDM-1 is capable of hydrolyzing all BLAs except for aztreonam, including also carboradienes known as the last line of antibiotics [48]. NDM-1 genes are mostly located in plasmids, which can be obtained by transposition and can easily spread horizontally among bacteria, and propagate rapidly all over the world [48]. Currently, the inhibition of NDM-1 is the main target of scientists’ research. As a result, a variety of molecules capable of inhibiting NDM-1 have been detected, including captopril, thiol compounds, Aspergillomarasmine A, new boric acid derivatives, sulfanilamide compounds, succinic acid derivatives, and a series of natural products [48]. Unfortunately, their unfavorable physicochemical properties and lack of specificity and safety to the human body still prevent their clinical use and no NDM-1 inhibitor has been commercialized to date [48]. The limited, and sometimes lack of, therapeutic options to counteract carbapenemase-producing bacteria, render the resistance to BLAs, particularly to carbapenems, one of the main current challenges in the healthcare systems worldwide [48].

## 3. β-Lactamase Enzymes Inhibitors (BLEsIs): What Works on Whom? 

One of the main approaches utilized to restore the efficacy of BLAs is to use BLEsIs. Except for those developed and clinically approved recently, BLEsIs used in therapy are molecules free of intrinsic antibacterial effects, which, by binding to the active site of the susceptible enzymes, prevent them from attacking and hydrolyzing the β-lactam ring of BLAs, thus extending the range of bacteria against which the antibiotics are effective [53,54]. 

Presently, BLEsIs are classified as inhibitors having a β-lactam core, and as non-β-lactam inhibitors having either a diazabicyclooctane (DBO) core or having other types of non-β-lactam cores. β-lactam inhibitors available on the market and approved for clinical use include clavulanic acid or clavulanate, sulbactam, and tazobactam, while the non-β-lactam ones include avibactam and relebactam having a DBO core and vaborbactam, which has a cyclic boronic acid core [55,56,57,58].

Particularly, among the clinically usable BLEsIs (six molecules), those of the first generation (clavulanate, sulbactam, and tazobactam) are effective against SBLEs and ESSBLEs of class A, not including carbapenemases, are weakly active on BLEs of class C, while they are completely inactive on MBLEs. Among the recently approved non-β-lactam BLEsIs, avibactam is effective against SBLEs and ESSBLEs of classes A and C, including carbapenemases and some clinically relevant carbapenemases of class D (OXAs), but not against MBLEs. On the contrary, relebactam and vaborbactam are mainly effective against SBLEs and ESSBLEs of classes A, C, and D and on KPCs (carbapenemases of class A) but are weakly active against OXAs (carbapenemases of class D) and are not active on MBLEs [31]. Importantly, all the six clinically approved BLEsIs are processed by SBLEs forming an initial covalent intermediate, deriving from a chemical interaction with the hydroxyl group of the serine present in the active site of the enzymes. Particularly, β-lactam BLEsIs such as clavulanate, sulbactam, and tazobactam bind to the serine like BLAs, but unlike BLAs, they form an irreversible bond with serine and act as suicide substrates leading to the inactivation of the enzyme [53]. On the contrary, non-β-lactam inhibitors, such as avibactam, relebactam, and vaborbactam bind reversibly to BLEs, and by a hydrolytic process, they reconstitute the original structure capable of inactivating new enzymes [53,59].

In addition to the currently marketed BLEIs, the following Table 3 reports new molecules that are still in preclinical or clinical study stages, or even the early stages of experimentation, and are not yet on the market.

The subsequent Table 4 describes the most common associations of BLEsIs present in Table 3 with existing antibiotics, the target BLE-producing bacteria, and other useful information concerning the route of administration and of excretion, and some pharmacokinetic data. While some of these combinations are already marketed and approved as therapeutics, while others are not and are only in preclinical/clinical trials or even in the in-vitro experimentation stage.

## 4. β-Lactam BLEsIs

β-lactam BLEsIs are “suicide inhibitors”, which irreversibly bind to the hydroxyl group of serine 70 in the active site of the enzyme forming an acyl derivative that undergoes secondary chemical reactions and relevant structural rearrangement, thus permanently inactivating the enzyme.

### 4.1. Clinically Approved β-Lactam BLEsIs

#### 4.1.1. Tebipenem 

Although not inserted in the Tables, because it is considered mainly an antibiotic, rather than a BLEsI, we make note of the existence of tebipenem, which is currently marketed only in Japan (brand name Orapenem) (Figure 3a) [4]. 

Particularly, tebipenem (C_16_H_21_N_3_O_4_S_2_, MW = 383.5), IUPAC name (4R,5S,6S)-3-[1-(4,5-dihydro-1,3-thiazol-2-yl)azetidin-3-yl]sulfanyl-6-[(1R)-1-hydroxyethyl]-4-methyl-7-oxo-1-azabicyclo[3.2.0]hept-2-ene-2-carboxylic acid, is a broad-spectrum carbapenem antibiotic, which was developed to replace antibiotics no longer active on bacteria that had acquired antibiotic resistance. Tebipenem is formulated as a prodrug in the form of tebipenem pivoxil ester (Figure 3b), for achieving better absorption in the gastrointestinal tract (GIT) and improved bioavailability. Tebipenem pivoxil is the first carbapenem orally administrable, which has proven good activity in treating ear infections, such as acute otitis media, and urinary tract infections sustained by MDR Gram-positive and Gram-negative organisms, in pediatric patients. In addition to working as BLAs, binding the PBPs, and inhibiting the synthesis of peptidoglycan, tebipenem is also a slow substrate that inhibits the β-lactamase enzymes produced by *Mycobacterium tuberculosis* [136,137,138]. Concerning the development and regulatory status, tebipenem is in phase III clinical trials in the United Kingdom and Europe, while in the US it is in the pre-registration phase [138].

#### 4.1.2. Clavulanic Acid (CA)

Clavulanic acid (C_8_H_9_NO_5_, MW: 199.16), IUPAC name, (2R,3Z,5R)-3-(2-hydroxyethylidene)-7-oxo-4-oxa-1-azabicyclo [3.2.0] heptane-2-carboxylic acid (Table 3) is a semi-synthetic BLA isolated from *Streptomyces clavuligerus*, whose biosynthesis has been recently reviewed [4]. As in penicillin, the four-term β-lactam ring of CA is condensed with a five-term cycle that differs from those of penicillin G and penicillin V because it is an oxazolidine (oxapenam) instead of a thiazolidine ring (penam). CA acts as reported previously. CA is currently combined with amoxicillin (2/1 ratio) and ticarcillin (30/1, 15/1, or 5/1 ratio) and details concerning the routes of administration and the related pharmacokinetic data are available in Table 4 [138,139,140,141,142,143,144]. The target bacteria of amoxicillin/CA are reported in Table 4. Particularly, amoxicillin/CA works against many Gram-positive species and against representatives of non-*Enterobacteriaceae* Gram-negative species, except for non-fermenting superbugs, such as *A. baumannii*, *A. pittii*, *A. nosocomialis*, *P. aeruginosa*, and *S. maltophilia*. Amoxicillin/CA has been used in the prophylaxis and treatment of polymicrobial infections, both in hospital and in primary care settings for 20 years, thanks to the broad-spectrum activity of such combinations, good tolerance, and therapy costs, which are lower than those required when other antimicrobial agents are used [143]. CA/amoxicillin is effective for treating diabetic foot infections, cIAIs, VAP, HAP, brain abscesses, cUTI, and to prevent infections after abdominal, pelvic, head, and neck surgery. Unfortunately, due to the excessive use of amoxicillin/CA, an increasing number of pathogens are developing resistance [145].

#### 4.1.3. Sulbactam

Sulbactam (C_8_H_11_NO_5_S, MW = 233.2), IUPAC name (2S,5R)-3,3-dimethyl-4,4,7-trioxo-4l6-thia-1-aza-bicyclo [3.2.0] heptane-2-carboxylic acid is a semi-synthetic BLEsI deriving from the (+)-6-aminopenicillanic acid [4], where the sulfur atom of the thiazole ring has been substituted by a sulfone group (Table 3). While free of any antibacterial activity against almost all bacteria, sulbactam is effective against bacteria belonging to the *Neisseriaceae* and *Acinetobacter* genera. Similar to CA, it is a suicide inhibitor of the SBLEs, thus preventing the inactivation of several BLAs [76,77,78]. Sulbactam is currently combined with ampicillin in ratio 2/1 or bounded to ampicillin to form a prodrug named sultamicillin [76,146,147,148] and with cefoperazone (1/1 and 1/2 ratio), a third-generation cephalosporin [149]. Details concerning the routes of administration and the related pharmacokinetic data for such combinations are available in Table 4 [76,146,147,148,149]. Sulbactam has a wider spectrum of action than CA, displaying activity against several ESSBLEs produced by Gram-positive and Gram-negative bacteria. Particularly, sulbactam is effective also on the ESSBLEs produced by the *Serratia*, *Providencia*, and *Morganella* genera. Sulbactam is capable of inhibiting BLEs produced by bacteria of the *Staphylococcus* and *Enterococcus* genera, as well as the plasmid-encoded ESSBLEs produced by *Enterobacteriaceae*, *H. influenzae*, *N. gonorrhoeae*, and *Moraxella* supp. While it inhibits also the chromosomal ESSBLEs produced by *P. mirabilis* and *B. fragilis*, sulbactam possesses weak activity on *Meningococcus*, *Gonococcus*, *Acinetobacter* genera, and *Burkholderia cepacia*.

Clinically, the ampicillin/sulbactam combination (2/1) is used to treat VAP, lower respiratory tract, and gynecological/obstetric infections, as well as cIAIs. Acute epiglottitis and periorbital cellulitis in pediatric subjects, diabetic foot infections, as well as skin and soft tissue infections are also treated with the same combination. While not active against *P. aeruginosa*, such an association works particularly well against infections caused by *A. baumannii*. Both parenteral and oral administration of ampicillin–sulbactam are suggested as therapy for treating several community-acquired infections in adults or children, including gynecologic infections from mild to moderate severity, or as an alternative for treating nosocomial infections sustained by carbapenem-resistant *A. baumannii* strains.

Since adverse effects are rare, the ampicillin–sulbactam association is still considered an efficacious defense against most adult and pediatric infections [150,151,152].

The combination of sulbactam/cefoperazone works better than cefoperazone alone against *Enterobacteriaceae*, *P. aeruginosa*, and *A. baumannii*. Interestingly, although weakly active on carbapenem-resistant *P. aeruginosa*, such a combination has presented higher in-vitro activity against ESSBLE-producing and AmpC-producing *Enterobacteriaceae*, as well as against carbapenem-resistant *A. baumannii*, even if sulbactam resistance was already observed in some *A. baumannii* strains [153]. Moreover, sulbactam does not inhibit KPC- and OXA-type carpapenemases, as well as MBLEs, regardless of the concentration used.

#### 4.1.4. Tazobactam

Tazobactam (C_10_H_12_N_4_O_5_S, MW = 300.3), chemically 3-methyl-4,4,7-trioxo-3-[1-3]triazol-1-ylmethyl-4l6-thia-1-aza-bicyclo [3.2.0] heptane-2-carboxylic acid, according to IUPAC, is a semi-synthetic BLEsI deriving from the (+)-6-aminopenicillanic acid [4], like sulbactam, where the sulfur atom of the thiazole ring has been substituted by a sulfone group. Additionally, with respect to the natural penicillanic acid, in the tazobactam structure, one of the exocyclic hydrogens of a methyl group has been replaced by a 1,2,3-triazol-1-yl group (Table 3) [65]. Tazobactam in the form of its sodium salt is combined with piperacillin (8/1 ratio) and ceftolozane (2/1) for the treatment of a variety of bacterial infections sustained by BLE-producing bacteria. Details concerning the routes of administration and the related pharmacokinetic data for both combinations are available in Table 4 [65].

Notably, piperacillin-tazobactam was initially approved by the FDA in 1994, while ceftolozane–tazobactam was approved by the FDA in 2014, providing wider antibacterial coverage for Gram-negative infections. In June 2019, ceftolozane–tazobactam was approved by the FDA for treating HAP and VAP, which are significant causes of morbidity and mortality in hospitalized patients [154]. 

Clinically, piperacillin/tazobactam is a broad-spectrum combination active against most Gram-positive and Gram-negative aerobic and anaerobic bacteria.

Administered to adults, piperacillin/tazobactam 8/1 is effective at treating lower respiratory tract, gynecologic, and skin/soft tissue infections, IAIs and UTIs. To treat patients affected by serious hospital-acquired infections, the co-administration of additional aminoglycoside antibiotics is suggested. While for the treatment of VAP, the co-administration of piperacillin/tazobactam with amikacin grants antimicrobial effects like those observed for the co-administration of ceftazidime and amikacin is suggested [155]. 

Additionally, the combination of piperacillin/tazobactam with amikacin is significantly more effective than that of ceftazidime with amikacin in the treatment of febrile episodes in patients with neutropenia or granulocytopenia complications [155].

When administered to treat IAIs, skin and soft tissue, or gynecologic infections, piperacillin/tazobactam determined outcomes like those evoked by standard aminoglycosides. Adverse events consist of gastrointestinal symptoms, such as diarrhea and skin reactions, whose frequency increases if piperacillin/tazobactam is administered in combination with an aminoglycoside [155].

The combination ceftolozane/tazobactam consists of a semi-synthetic broad-spectrum fifth-generation cephalosporin structurally like ceftazidime (ceftolozane), with tazobactam [156]. Such a combination is active against ESSBLEs produced by *Enterobacteriaceae* and MDR *P. aeruginosa*, and it was recently approved for the treatment of cIAIs and cUTIs [157]. Unfortunately, the emergence of resistance to ceftolozane/tazobactam has been reported recently [158].

### 4.2. β-Lactam BLEsIs in Experimental Phase 

Unfortunately, among the β-lactam BLEsIs reported in Section 4.1 and already approved for clinical use, none are capable of inhibiting carbapenemases. In recent years, within the group of penicillin sulfones, several molecules have emerged with higher or broader activity than the traditional inhibitors mentioned above, among which enmetazobactam and LN-1-255 have exhibited relevant activity against carbapenemases.

#### 4.2.1. Enmetazobactam

Enmetazobactam, formerly AAI101 (C_11_H_14_N_4_O_5_S, MW = 314.3), chemically (2S,3S,5R)-3-methyl-3-[(3-methyltriazol-3-ium-1-yl)-methyl]-4,4,7-trioxo-4λ^6^-thia-1-azabicyclo [3.2.0] heptane-2-carboxylate, according to IUPAC, is one of the novel compounds synthesized using penicillanic acid as the template molecule [4] currently under clinical trials. Particularly, being a penicillanic acid sulfone, enmetazobactam is structurally like tazobactam, except for the presence of a methyl group in the triazole ring, which confers the compound a net neutral charge capable of promoting its entry into the bacterial cell, thus enhancing its activity [66]. Enmetazobactam inhibits mainly class-A ESSBLEs, including KPCs, such as KPC-2 and KPC-3 with nanomolar IC_50_ values similarly potent to avibactam (see later), while displaying only weak activity against class-C and -D carbapenemases, being significantly less active than avibactam [66,67]. Enmetazobactam was paired by Allecra Therapeutics with cefepime (ratio 4/1), a fourth-generation cephalosporin, to achieve an investigational combination, currently in clinical trials. Recently, it was announced that such a combination met the European Medicines Agency and US FDA pre-specified primary endpoint in the phase 3 ALLIUM clinical trial, in which it was tested in the treatment of cUTIs, or acute pyelonephritis [68]. 

The combination showed strong activity against *Enterobacteriaceae* that produce class-A ESSBLEs, including some bacteria producing class-A and class-D carbapenemases, such as KPCs and OXAs, respectively [69,70]. Particularly, in infections models due to cefepime-resistant *Enterobacteriaceae*, the use of the cefepime-enmetazobactam combination significantly improved the infections conditions [71]. Furthermore, the association cefepime/enmetazobactam showed better activity and comparable safety and tolerability against MDR ESSBLE-producing *Enterobacteriaceae* than the recently approved and already marketed BLEsIs/BLAs combinations including ceftazidime/avibactam, ceftolozane/tazobactam, and imipenem/relebactam [70]. In this scenario, it seems that Allecra Therapeutics is thinking of developing an additional combination of enmetazobactam with piperacillin.

#### 4.2.2. 6-Methylidene Penems

6-methylidene penems are penem-like molecules with various substitutions at the C6 position via a methylidene linkage. Particularly, penem molecules with heterocycle substitutions were investigated for their activities and efficacy as beta-lactamase inhibitors, evidencing that these molecules resulted in 50% inhibition of enzyme activity at concentrations of 0.4 to 3.1 nM for the TEM-1 enzyme, 7.8 to 72 nM for Imi-1, 1.5 to 4.8 nM for AmpC, and 14 to 260 nM for a CcrA metalloenzyme [72]. The chemical compound having the IUPAC name 6-(6,7-dihydro-5H-cyclopenta[d]imidazo [2,1-b]thiazol-3-ylmethylene)-7-oxo-4-thia-1-aza-bicyclo [3.2.0] hept-2-ene-2-carboxylate, sodium salt (C_15_H_10_N_3_NaO_3_S_2_, MW = 367.0) (Figure 4), is an example of 6-methylidene penem that displayed activity like BLEsIs.

When these compounds were co-administered with piperacillin 1/1 in-vitro studies, the piperacillin MICs were reduced 2–32-fold against ESSBLE-producing *E. coli* and *K. pneumoniae* strains.

Furthermore, the MICs against piperacillin-resistant (MIC of piperacillin > 64 μg/mL) strains belonging to *Enterobacter* spp., *Citrobacter* spp., and *Serratia* spp. were recovered to the level of susceptibility (MIC of piperacillin, ≤16 μg/mL) when piperacillin was combined with the 6-methylidene penems in ratios of 4/1, 8/1, or 16/1. Preclinical studies on mice confirmed the ability of these combinations of penem-type inhibitors/piperacillin to protect mice from acute lethal bacterial infections due to class-A and -C β-lactamase and extended-spectrum BLE-producing pathogens [72]. Pharmacokinetic parameters were determined and a few details are reported in Table 4. Median effective doses were reduced approximately two- to eight-fold compared to those of piperacillin alone when the drug was combined with the various inhibitors at a 4:1 ratio. 

#### 4.2.3. LN-1-255 

LN-1-255 (C_22_H_19_N_2_NaO_9_S, MW = 510.4), IUPAC name (2S,3R,5R,6Z)-4-thia-1-azabicyclo [3.2.0] heptane-2-carboxylic acid, 3-[[[(3,4-dihydroxyphenyl) acetyl] oxy] methyl]-3-methyl-7-oxo-6-(2-pyridinylmethylene)-4,4-dioxide, monosodium salt, is a 6-alkylidene-2’-substituted penicillin sulfone [4]. Like tazobactam and enmetazobactam, in the LN-1-255 structure, one hydrogen atom of a methyl group has been substituted, in this case, with a 3,4-dihdroxyphenyl acetate ester group. Additionally, in L-1-255 a 2-pyridinylmethylene group has been added in the α position to the carbonyl of the β-lactam ring. LN-1-255 acts as an effective ESSBLEsI, which proved to be capable of inhibiting the BLEs of class A as SHV-1 and SHV-2 with Ki of 110 nM and 100 nM, respectively. Also, it was demonstrated to inhibit the class-D carbapenemase OXA-48 produced by *Enterobacteriaceae* with IC_50_ of 3 nM [73]. It was reported, that LN-1-255 presented an affinity in the nM range for several CHDBLEs produced by *A. baumannii*, including the relevant OXA-23, OXA-24/40, OXA-58, and OXA-143 [74]. By combining carbapenems with LN-1-255, an important decrease in the MICs of these antibiotics on carbapenase-producing *A. baumannii* has been observed [74]. Successful preclinical trials in mice have been performed to evaluate a new therapy consisting of the use of a combination of imipenem and LN-1-255 for the treatment of an experimental murine pneumonia model caused by carbapenem-resistant *A. baumannii* transformants and clinical isolates carrying CHDBLEs [73]. Details concerning the administration routes adopted and the pharmacokinetic parameters that were determined are reported in Table 4. No toxicity was observed in treated mice. It has been reported that the presence of the catechol group is responsible for the effectiveness of LN-1-255. Particularly, the catechol group would promote the internalization of the molecule in the bacteria by an iron uptake system [31]. Recently, some 6-arylmethylidene penicillin-based sulfones obtained modifying the pyridine ring of LN-1-255 proved improved efficiency against CHDBLEs [75].

#### 4.2.4. Other Synthetized β-Lactam BLEsIs

A series of cephalosporin-derived reverse hydroxamates and oximes were prepared, and in-vitro evaluated as inhibitors of VIM-2 and GIM-1, belonging to the family of MBLEs. Additionally, they were also kinetically evaluated as substrates of some SBLEs produced by *E. cloacae* including P99 (class C), TEM-1 (class A), and OXA-1 (class D) [159]. Particularly, the cephalosporin-derived reverse hydroxamates, corresponding to the IUPAC name 3-acetoxymethyl-7-(hydroxy-phenylacetyl-amino)-8-oxo-5-thia-1-aza-bicyclo [4.2.0] oct-2-ene-2-carboxylate, sodium salt (Figure 5) were shown to inhibit the GIM-1 MBLE at a submicromolar concentration (IC_50_ = 300 nM) [159]. 

With respect to the class-A, -C, and -D SBLEs, all the reverse hydroxamates showed low K_m_ values, thus indicating that they were recognized in a manner similar to the non-hydroxylated N–H amide side chains of the natural substrates of these enzymes. However, the K_cat_ values displayed by the reverse hydroxamates were up to three orders of magnitude lower than those of the natural substrates, thus evidencing a significant reductive action of the hydrolytic activity of P99, TEM-1, and OXA-1 SBLEs. Although the replacement of the amide N–H bond in the natural substrates with N–OH may represent a useful strategy for the inhibition of serine hydrolases the degree of inactivation of these enzymes is not sufficient to be clinically useful.

## 5. Non-β-Lactam BLEsIs

### 5.1. Diazabicyclooctanes (DBOs)

Among the new families of non-β-lactam BLEsIs, DBOs consist of molecules that do not possess a β-lactam ring but have a five-membered DBO ring and an amide group in common, which is responsible for the interaction with the active-site of SBLEs via a carbamylation reaction [67]. Notably, while the β-lactam BLEsIs, because of the irreversible interaction with the enzymes, go through the opening of the ring to form an acyl derivative and then undergo relevant and irrecoverable structural rearrangement, DBOs do not. Particularly, DBOs bind reversibly to the enzyme in the active site, undergo the opening of the DBO ring forming carbamate derivatives that, following a de-acylation process, regenerate the original compounds, which can inactivate new enzymes.

Thanks to this singular mechanism of action, DBOs display high efficiency on class-A and C-SBLEs, while variable effects against class-D enzymes [67].

#### 5.1.1. Clinically Approved DBOs

##### Avibactam 

Avibactam (C_7_H_11_N_3_O_6_S, MW = 265.3), IUPAC name [(2S,5R)-2-carbamoyl-7-oxo-1,6-diazabicyclo[3.2.1]octan-6-yl] hydrogen sulfate consists of a DBO ring in which the hydrogen on the nitrogen atom at position 6 has been replaced by a sulfonic acid group that can provide easily sodium salts [76]. 

Avibactam possesses a broad spectrum of activity at nanomolar concentrations against the most clinically relevant ESSBLEs of classes A, C, and D, such as KPCs, GESs, CTX-Ms, SHVs, the chromosomal AmpCs of *P. aeruginosa* and *E. cloacae*, and, even if at a less significant level, OXA-48 [77,78].

The combination with ceftazidime (ratio 4/1), a third-generation cephalosporin antibiotic, was approved by the FDA on 25 February 2015 and it is currently administered also in Europe for the treatment of cIAIs (in combination with metronidazole), the treatment of cUTIs including pyelonephritis, and HAP or VAP caused by antibiotic-resistant pathogens and MDR Gram-positive and Gram-negative bacteria [76]. Based on data concerning its clinical safety, avibactam should be reserved for patients over 18 years old who have limited or no alternative treatment options. Evaluations in patients with cystic fibrosis and pediatric patients with HAP are currently underway [67]. Despite being among those recently approved, the ceftazidime/avibactam combination showing the broadest therapeutic spectrum remains inactive against MBLEs and the most ESSBLEs of OXA type. The co-administration of avibactam re-established the activity of ceftazidime against KPCs, OXA-48, and ESSBLEs produced by *Enterobacteriaceae* and by β-lactam resistant isolates of *P. aeruginosa*, which overexpress AmpC and are OprD deficient [67]. Details concerning the routes of administration and the related pharmacokinetic data are available in Table 4 [76]. Avibactam is mainly excreted with urine and consequently, in subjects affected by renal impairment, it is advisable to tune the dosage, as renal dysfunctions might change the pharmacokinetics of avibactam. Additionally, the co-administration of drugs that can influence the renal elimination of avibactam should be precluded or checked to avoid negative impact on the pharmacokinetics of avibactam [76]. The tolerability of ceftazidime–avibactam is like that of ceftazidime alone [79], but unfortunately, cases of resistance to this initially successful combination have already been described [158]. In recent years, new combinations of BLEsIs/BLAs were developed to extend the spectrum of action DBOs. In this context, it was supposed to be the association of avibactam, which does not exert inhibitory activity against MBLEs, with aztreonam, which is the only monobactam approved for clinical uses. Since aztreonam is stable with MBLEs but needs protection from ESSBLEs or AmpC enzymes, inactivated by avibactam, its combination with avibactam would allow one to obtain a therapeutic mixture active also on bacteria that produce MBLEs [80]. Accordingly, a new combination of avibactam with aztreonam, which proved to be a promising curative option against MBLE-producing *Enterobacteriaceae*, is in development [81,160]. Although both avibactam and aztreonam have already been accepted for clinical use, and the efficacy and safety of both compounds are well established, concerning their combination, further investigations are necessary to assess their combinative activity against MBLE-producing *P. aeruginosa* [161].

Successful clinical cases treated with the novel combination have already been reported [162,163,164], and two studies in clinical phase III are currently in progress to assess its effectiveness, tolerability, and safety in the treatment of severe infection by MBLE-producing Gram-negative bacteria.

##### Relebactam

Like avibactam, relebactam (C_12_H_20_N_4_O_6_S, MW = 348.4, IUPAC name [(2S,5R)-7-oxo-2-(piperidin-4-ylcarbamoyl)-1,6-diazabicyclo[3.2.1]octan-6-yl] hydrogen sulfate is a DBO compound [4]. It differs from avibactam in the presence of a basic piperidine ring attached to the nitrogen atom of the acyl amide group, which produces a positive charge at physiological pH, able to decrease its export from the bacterial cell, thus enhancing its inhibiting activity. Like avibactam, relebactam does not incur degradation but has a slightly closer spectrum of inhibition than avibactam. In fact, relebactam does not display in-vitro activity against OXA-48 enzymes and possesses less of a capability to block some class-A enzymes such as CTX-Ms and KPCs. In many cases, relebactam has demonstrated, in vitro, strong inhibition activity, comparable to that of avibactam against most class-A and class-C BLEs [67].

Currently, relebactam is marketed and administered in combination with imipenem and cilastatin to restore their activity against BLE-producing bacteria, as a treatment for cUTIs, pyelonephritis, and cIAIs in adults. 

Approved by the FDA in July 2019, the imipenem–cilastatin/relebactam (ICR) association (ratios 2/2/1), now available on the market with the brand name of Recarbrio^®^, is considered a last-line treatment option [83]. Phase 3 clinical trials evidenced that ICR may be used also to treat HAP and VAP caused by bacteria resistant to imipenem and atypical mycobacterial infections. Details concerning the routes of administration and the related pharmacokinetic data for such combination are available in Table 4 [83].

Particularly, in in-vitro studies, ICR was active on several MDR pathogens, including carbapenem-resistant *P. aeruginosa*, *Enterobacteriaceae* producing ESSBLEs and resistant to carbapenems and the anaerobic *Bacteroides* spp. Particularly, ICR showed high activity against XDR *P. aeruginosa*, including strains resistant to imipenem due to OprD deficiency and strains with acquired resistance to ceftolozane/tazobactam and ceftazidime/avibactam, due to the production of GES-1, PER-1, and extended-spectrum OXA enzymes [67]. Importantly, ICR exhibited strong activity against several isolates of *Enterobacteriaceae*, including those species resistant to all classical BLAs and fluoroquinolones, those producing KPCs and ESSBLEs, those with porin alterations, and against international clones of *K. pneumoniae* and *E. coli* [67]. ICR was scarcely active on Gram-negative bacteria producing OXAs, and totally inactive against MBLE-producing strains [13]. Notably, ICR was inactive against isolates of *A. baumannii* producing horizontally acquired OXA-23, OXA-24/40, and OXA-58 carbapenemases. 

Nausea (6%), diarrhea (6%), and headache (4%) are the main adverse reactions observed [84,85].

Cases of bacterial resistance to ICR combination have been recently described [158].

#### 5.1.2. DBOs in Experimental Phase

In recent years, avibactam and relebactam have been taken as template molecules on which to make structural changes to prepare compounds with improved activity [67].

To this end, second-generation DBO-type inhibitors have been developed, including durlobactam, zidebactam, and nacubactam. Differently from most BLEsIs including previous DBOs, which are free from intrinsic antibacterial effects, these new generation DBOs possess a high affinity for the PBP2 of many Gram-negative species, thus exerting antimicrobial effects by impeding the formation of the bacterial cell wall, like BLAs. When combined with BLAs, the new DBOs are capable of inhibiting carbapenemase-producing pathogens, including bacteria that produce those of class B.

##### Zidebactam (WCK-5107)

Zidebactam (C_13_H_21_N_5_O_7_S, MW = 391.4), IUPAC name [(2S,5R)-7-oxo-2-[[[(3R)-piperidine-3-carbonyl]amino]carbamoyl]-1,6-diazabicyclo [3.2.1] octan-6-yl] hydrogen sulfate belongs to the new generation of DBOs [86,87]. Zidebactam, in addition to possessing inhibitory activity on ESSBLEs like that of avibactam and relebactam, possesses intrinsic antibacterial effects. Zidebactam, due to its affinity for PBPs, binds them, thus inhibiting the synthesis of peptidoglycan and the formation of the bacterial cell wall [67]. Zidebactam, when co-administered with BLAs while protecting them with BLEs activity, also displays synergistic antibacterial effects with the associated antibiotics. Structurally, it differs from avibactam and relebactam which have an acyl amide group on the DBO nucleus, for having an acyl hydrazide moiety. On the other hand, like relebactam, it bears a basic piperidine residue, particularly an acyl piperidine group, which produces a positive charge at physiological pH, thus enhancing its activity [67,86]. Zidebactam displayed activity against Gram-negative bacteria such as *Enterobacteriaceae*, and non-fermenting species such as *P. aeruginosa* and *A. baumannii*, including strains producing BLEs of classes A, C, and D, such as CTX-M-like, AmpCs or OXA-48-like enzymes [67]. Unlike avibactam, which displayed a low affinity for PBP2 of *P. aeruginosa*, thus limiting its activity to *Enterobacteriaceae*, Zidebactam has a high affinity for PBP2, thus extending the spectrum of action of previous DBOs to non-fermenting bacteria of Gram-negative species [88]. Currently, Zidebactam is combined with cefepime in dosage forms suitable for parenteral administration and is being tested to counteract severe infections by MDR Gram-negative bacteria. Notably, the combination is presently under investigation in several phase I clinical trials, to assess the safety, tolerability, and pharmacokinetics in adults [23,67,87]. Interestingly, while zidebactam alone displayed very low MICs against most *E. coli*, *K. pneumoniae*, *Citrobacter* spp., and *Enterobacter* spp., some representative isolates of these same species, most *Serratia* spp., and species belonging to the *Proteaceae* family turned out to be resistant (MIC > 32 µg/mL) [23]. Furthermore, in the case of zidebactam-resistant isolates producing class-A and class-C enzymes, the co-administration of zidebactam with cefepime translated into an improvement of cefepime activity thus confirming the potent activity of zidebactam as an inhibitor of ESSBLEs. The combination cefepime/zidebactam was effective against almost all *Enterobacteriaceae* producing AmpCs, ESSBLEs, K1, KPCs, and OXA-48-like BLEs. On *P. aeruginosa* isolates producing AmpCs and MBLEs, cefepime/zidebactam displayed an MIC of 8 mg/L. While, the combination was successful on *S. maltophilia* strains, it provided poor results on *A. baumannii* isolates [23]. Cefepime–zidebactam inhibited 98.5% carbapenem-resistant *Enterobacteriaceae* at ≤8 µg/mL, 94.9% *Enterobacteriaceae* producing MBLEs at ≤8 µg/mL, and 99.6% MDR *P. aeruginosa* isolates at ≤32 µg/mL, including all MBLE-positive isolates [89]. Cefepime–zidebactam was successful on most isolates of *Enterobacteriaceae* and *P. aeruginosa* isolates on which ceftazidime–avibactam, ceftolozane–tazobactam, imipenem–relebactam, and colistin failed [23,89]. Isolates of *S. maltophilia* and *Burkholderia* spp. were inhibited by cefepime–zidebactam at ≤32 µg/mL [89].

Overall, in a clinical trial registered at ClinicalTrials.gov under identifier NCT02942810, the cefepime/zidebactam combination was found to be safe and well-tolerated both in subjects with normal and impaired renal function [89]. Particularly, the effects of renal impairment on the pharmacokinetics and safety of the combination were investigated in 48 subjects based on Cockcroft–Gault-estimated creatinine clearance (CL_CR_) [89]. Details concerning the routes of administration and the related pharmacokinetic data for cefepime-zidebactam in healthy subjects are available in Table 4 [89].

##### Durlobactam (ETX-2514)

Durlobactam (C_8_H_11_N_3_O_6_S, MW = 277.36), IUPAC name, [(2S,5R)-2-carbamoyl-3-methyl-7-oxo-1,6-diazabicyclo [3.2.1] oct-3-en-6-yl] hydrogen sulfate, differently from the previous clinically approved and not yet approved DBOs, is a diazabicyclo-octenone having a double bond in the seven-termed ring and a methyl group on one of the carbon atoms forming the double bond. For the rest, it is structured as avibactam [90]. Durlobactam inhibits BLEs of classes A, C, and D [91] and possesses intrinsic antimicrobial activity against some *Enterobacteriaceae* due to the capability to bind PBPs [92,165]. To limit the increasing resistance of *A. baumanni* to sulbactam, due to the emergence of OXAs (carbapenemases of class D) and AmpCs, durlobactam is being tested in combination with sulbactam, currently marketed in combination with ampicillin and capable of inhibiting only class-A and -C BLEs, not including carbapenemases [67]. The developed durlobactam/sulbactam combination was successful in counteracting infections caused by MDR isolates of *A. baumannii*, including those producing OXA-type carbapanemase [67]. While capable of restoring the efficacy of imipenem against isogenic *P. aeruginosa* strains overexpressing carbapenemases such as KPCs and OXA-48 [91], durlobactam was not active against class B carbapenemases such as NDMs [67]. Durlobactam has already been evaluated in clinical trials as a new option for treating infections such as bacteremia, HAP, and VAP caused by the *A. baumannii-calcoaceticus* complex (ABC) including strains of ABC resistant to colistin, as well as to cure acute pyelonephritis (APN) and cUTIs sustained by the same pathogen. Currently, durlobactam is being studied in phase III clinical trial NCT03894046 to evaluate the efficacy and intravenous safety in combination with imipenem and sulbactam in the treatment of patients with infections caused by ABC [90,93]. Particularly, the phase II clinical trial described by Sagan et al. evidenced that the combination imipenem/cilastatin/sulbactam/durlobactam 1/1/1/1 was generally well-tolerated in moderately ill, hospitalized adults with cUTIs or APN [93]. Based on the pharmacokinetic parameters established in the study, the optimal target success against members of the ABC was achieved with a sulbactam/durlobactam dose of 1000 mg (of each component) administered every 6 h (q6h) via a 3-h intravenous (i.v.) infusion (Table 4) [93]. 

##### Nacubactam (OP-0595)

Nacubactam (C_8_H_16_N_4_O_7_S, MW = 324.3), IUPAC name [(2*S*,5*R*)-2-(2-aminoethoxycarbamoyl)-7-oxo-1,6-diazabicyclo [3.2.1] octan-6-yl] hydrogen sulfate, like zidebactam and dorlebactam, acts both as an inhibitor of SBLEs of classes A, C, and some of D and as an antibiotic by inhibiting PBP2, thus both protecting the co-administered drug by hydrolysis and enhancing its activity [166]. Structurally, it differs from avibactam by an ethanolamine residue bound to the nitrogen atom of the acyl amine through its oxygen atom. Hence, nacubactam possesses a primary amine group that produces a positive charge at physiological pH, thus enhancing its activity, with the mechanism previously described for relebactam [94]. Though it acts as a potent inhibitor of class A carbapenemases including KPCs, nacubactam has a low affinity for those of class D such as OXA-23 and OXA-24/40, while data concerning OXA-48 are lacking. On the contrary, it possesses the highest activity against TEMs and CTX-Ms [67]. 

Currently, nacubactam is under investigation associated with either meropenem or cefepime in dosage forms for intravenous administration. In investigations concerning the association nacubactam/meropenem, good synergy against carbapenem-resistant KPC-producing *Enterobacteriaceae* was observed [94,166], while both the associations meropenem/nacubactam and cefepime/nacubactam were capable of inhibiting 80.3% and 93.3% of MBLE-producing *Enterobacteriaceae*, respectively. Applying the same therapeutic strategy thought for avibactam, nacubactam was tested also in association with aztreonam, which is a monobactam antibiotic resistant to MBLEs, affording a drug formulation almost universally active against the MBLE-producing *Enterobacteriaceae* [95]. While in-vitro studies have established the inability of nacubactam/meropenem to inhibit meropenem-resistant *P. aeruginosa* isolates [96], when tested in vivo on models of infections caused by meropenem-resistant KPC-producing strains of *P. aeruginosa*, the combination has exhibited promising antibacterial activity, thus unveiling a potential role of this drugs combination in the treatment of infections caused by this species [97]. However, nacubatam as an antibiotic was completely inactive against *A. baumannii* [96], as confirmed by recent studies that reported that, on this specie, nacubactam acts only as an inhibitor of the BLEs produced [167]. Two phase I clinical trials have been completed to determine the intrapulmonary concentrations of nacubactam and meropenem and to study the safety, tolerability, and pharmacokinetics in healthy volunteers (Clinicaltrial.gov identifier: NCT02134834 and NCT03182504). Particularly, the placebo-controlled clinical trials investigating the safety, tolerability, and pharmacokinetics of intravenous nacubactam were recently described (identifiers NCT02134834, single ascending dose study, and NCT02972255, multiple ascending dose study), (Table 4) [98]. Notably, healthy participants received single ascending doses of nacubactam (50–8000 mg), multiple ascending doses of nacubactam (1000–4000 mg, every 8 h for up to 7 days), or nacubactam 2000 mg associated with meropenem 2000 mg, q8h for 6 days after a 3-day lead-in period. Nacubactam was generally well-tolerated, and the most frequently reported adverse events (AEs) were mild to moderate complications associated with intravenous access and headache. Nacubactam was excreted largely unchanged into urine. Co-administration of nacubactam with meropenem did not significantly alter the pharmacokinetics of either drug. These findings support the continued clinical development of nacubactam and demonstrate the suitability of meropenem as a potential β-lactam partner for nacubactam. Currently, nacubactam is being studied in phase I clinical trial NCT03174795 for the evaluation of the pharmacokinetics of nacubactam and meropenem in participants with cUTIs [94].

##### ETX-1317

ETX-1317 (C_10_H_12_FN_3_O_5_, MW = 295.2), IUPAC name (2R)-2-(((2S,5R)-2-carbamoyl-3-methyl-7-oxo-1,6-diazabicyclo [3.2.1] oct-3-en-6-yl)oxy)-2-fluoroacetic acid is a new DBO which proved to possess broad-spectrum activity against class-A and class-C ESSBLEs. Structurally, ETX-1317 resembles durlobactam with the presence of a double bond bringing a methyl group in the seven-termed ring, but differently from all other DBOs, it does not contain the sulfonic acid group, which has been replaced by a residue of 2-fluoroacetic acid linked to the nitrogen atom of the cyclic urea via an oxygen bridge. In the form of its isopropyl ester prodrug ETX-0282 (C_13_H_18_FN_3_O_5_, MW = 315.3), IUPAC name isopropyl (R)-2-(((1R,2R,5R)-2-carbamoyl-4-methyl-7-oxo-1,6-diazabicyclo[3.2.1]oct-3-en-6-yl)oxy)-2-fluoroacetate ETX-1317 is orally administrable [99,100]. Currently, Entasis (Boston BioHub, Waltham, MA USA) is developing ETX0282 in combination with cefpodoxime proxetil, an orally available third-generation cephalosporin approved for the treatment of a variety of bacterial infections [101] encompassing severe infections by MDR and carbapenem-resistant *Enterobacteriaceae*, including KPC-producing species [67]. Note that, while DBOs are particularly suitable for intravenous administration but poorly available if oral administered, the administration of the prodrug ETX-0282 allowed good oral dosing availability [101]. A phase I clinical trial (Clinicaltrial.gov identifier: NCT03491748) to evaluate the safety, tolerability, and pharmacokinetics of ETX-1317 when administered orally to healthy individuals has recently been completed [101]. Like other new generation DBOs, ETX-1317 was also proved to have an affinity for the PBP2 of *E. coli*, but its intrinsic antimicrobial activity was lower than that of zidebactam or durlobactam. However, the MICs of cefpodoxime/ETX0282 combination against several isolates of *Enterobacteriaceae* including also those producing KPCs, OXA-48, or MBLEs carbapenemases were very low (0.013 mg/L) [102].

Table 4 reports some pharmacokinetic data obtained from preclinical trials in which the combination cefpodoxime proxetil/ETX0282 was orally administered [103] or dispensed by intravenous infusion [168].

Reports on the antibacterial properties of cefpodoxime/ET1317 on other MDR pathogens such as non-fermenting isolates of *P. aeruginosa* or *A. baumannii* were not found.

##### WCK-5153

WCK-5153 (C_12_H_19_N_5_O_7_S, MW = 377.1), IUPAC name [(2S,5R)-7-oxo-2-(2-((S)-pyrrolidine-3-carbonyl)hydrazine-1-carbonyl)-1,6-diazabicyclo [3.2.1] octan-6-yl] hydrogen sulfate, is a new inhibitor of PBP2, structurally like zidebactam with an acyl hydrazide group externally acylated with an acyl pyrrolidine in place of an acyl piperidine [104]. It has been demonstrated by mass spectroscopy, that WCK-5153 inhibits SBLEs of classes A and C, and has a particular affinity for AmpCs, by forming highly stable reversible acyl-complexes, while it is inactive against the tested OXAs [105]. Additionally, crystallography revealed that it forms complexes with KPC-2 adopting a “chair conformation” with the sulfate occupying the carboxylate binding region [105]. WCK-5153 showed to remarkably potentiate the activity of co-administered BLAs against *P. aeruginosa*, including MDR high-risk clones that produce MBLEs. Also, WCK-5153 demonstrated a specific high affinity for PBP2 of *A. baumannii*, acting as a potent β-lactam enhancer against isolates of this species [104]. No clinical trials have yet been conducted with this compound.

##### WCK-4234

WCK-4234 (C_7_H_8_N_3_NaO_5_S, MW = 269.3), IUPAC name [(2S,5R)-2-cyano-7-oxo-1,6-diazabicyclo [3.2.1] octan-6-yl] sulfate sodium salt is a novel potent BLEsI. Structurally, WCK-4234 is like avibactam with a nitrile group in place of the carbamide residue [106]. It was thought that the nitrile group could be advantageous in reducing the steric bulk of the R1 substituents present in the previously reported DBOs, thus yielding a better fit into sterically constrained active sites of OXAs (class D) [105]. WCK-4234 inhibited SBLEs of classes A, C, and D with unprecedented K_2_/K values against OXA-type carbapenemases [105]. Studies using mass spectrometry evidenced that WCK-4234 forms highly stable acyl-complexes both with enzymes of class A such as KPC-type carbapanemases, of class C (AmpCs), and of class D (OXAs). Particularly, mass spectroscopy demonstrated that WCK-4234 inhibits SBLEs by forming highly stable reversible acyl-complexes, while crystallography revealed that it forms complexes adopting a “chair conformation” with the sulfate occupying the carboxylate binding region [105]. Usually, DBOs are active on SBLEs of classes A and class C, do not inhibit MBLEs, and display variable behavior against class-D carbapenemases, but generally do not work against OXA-type carbapenemases of *Acinetobacter* spp. [107]. Unusually, WCK-4234 proved to be a potent inhibitor of class-A and class-D carbapenemases, as well as of enzymes of class C. WCK-4234 demonstrated potent cross-class inhibition and further clinical studies targeting MDR infections are reasonable.

Even if free of any intrinsic antibacterial effect, WCK-4234 was capable of enhancing the activity of carbapenems against *Enterobacteriaceae* producing OXA-48 or KPC carbapenemases but not against *Enterobacteriaceae* producing MBLEs [67]. Importantly, while combined with meropenem, WCK-4234 displayed activity against OXA-23, OXA-24/40, and OXA-51 hyperproduced by *A. baumannii*. Unfortunately, the same combination did not work against a collection of carbapenem-resistant isolates of *P. aeruginosa* [67].

No clinical trials have yet been conducted with this compound. On the contrary preclinical trials in murine models of neutropenic lung infection proved that the combination meropenem-cilastatin-WCK-4234 decreased by 2.5 log the bacterial load of MDR isolates of *A. baumanni* producing OXA-23 (Table 4) [105].

##### ANT-3310

ANT-3310 (C_6_H_9_FN_2_O_5_S, MW = 240.2), IUPAC name [(2R,5R)-2-fluoro-7-oxo-1,6-diazabicyclo [3.2.1] octan-6-yl] hydrogen sulfate is a novel broad-spectrum serine β-lactamase inhibitor of the DBO class, free of intrinsic antibacterial activity. Structurally, ANT-3310 is like WCK-4234 with a fluorine atom in place of the nitrile group [108]. It is capable of potentiating the activity of meropenem against carbapenem-resistant *Enterobacteriaceae* (CRE) and carbapenem-resistant *A. baumannii* (CRAB), which are WHO priority pathogens [108,109,110]. Particularly, it displayed excellent inhibitory activity against both KPC-like and OXA-like carbapenemases.

Recently, the US FDA has granted qualified infectious disease product (QIDP) designation to the meropenem (MEM)-ANT-3310 combination developed by Antabio SAS. Particularly, QIDP was granted to MEM-ANT-3310 for the treatment of cUTIs, HAP, VAP, and cIAIs [109].

Currently, in preclinical development, ANT-3310 will be administered intravenously in conjunction with meropenem, for treating hospital-acquired infections caused by Gram-negative pathogens [109]. Notably, ANT-3310 was capable of potentiating meropenem activity against CRAB in murine infection models and restoring meropenem susceptibility in 95% of *A. baumannii* clinical isolates. Additional details concerning this study are reported in Table 4. Since MEM-ANT-3310 targets both CRE and CRAB, as well as is active against most isolates of *P. aeruginosa*, this combination’s results are largely different from other combinations of BLAs/BLEIs, in terms of broad activity against alarming MDR Gram-negative pathogens, responsible for severe nosocomial infections. The activity against both CRAB and CRE provides the potential for MEM-ANT3310 to be an important addition to the available therapeutic armamentarium [109].

##### GT-055 (also Referred to as LCB18-055) 

GT-055 (C_13_H_20_F_3_N_5_O_8_S, MW = 463.4), chemical name sulfuric acid mono-[2-(5,5-bis-aminomethyl-4,5-dihydro-isoxazol-3-yl)-7-oxo-1,6-diaza-bicyclo [3.2.1] oct-6-yl] ester; compound with trifluoro-acetic acid, is a novel potent broad-spectrum BLEI of DBO family [111]. Differently from the last two novel DBOs herein reported, which have residues less bulky than the carbamide or carbo-hydrazine residues of other BODs, GT-055 possesses an isoxazole ring, which in turn binds to the carbon 5, two ethylamine residues that provide positive charges at physiological pH [111]. GT-055 exerted intrinsic antibacterial activity against many isolates of *Enterobacteriaceae*, including *E. coli* and *K. pneumoniae* by binding to PBP2 and inhibiting the formation of the bacterial wall [112]. Currently, it is under investigation in combination with GT-1 (also known as LCB10-0200), a novel siderophore-di-hydroxy-pyridone conjugated with an amino-thiazolyl-glycyl-modified cephalosporin. GT-1 works by exploiting the bacterial iron-uptake systems, which promote its entry into Gram-negative pathogens using a “Trojan-horse” strategy.

Particularly, GT-1 exhibited in-vitro activity (≤2 μg/mL MICs) against many MDR isolates, including *Enterobacteriaceae* capable of producing ESSBLEs and carbapenemases, such as *E. coli*, *K. pneumoniae*, and the non-fermenting *Acinetobacter* spp. producing OXA-like enzymes. Less powerful than GT-1, GT-055 was active only against isolates of *E. coli* and *K. pneumoniae* (MIC 2–8 μg/mL), but in combination with GT-1, GT-055 was able to enhance its activity against many GT-1–resistant strains [112].

Being only in its early stages of development, it is not yet clear whether GT-055 is capable of inhibiting carbapenemases, but the synergic action between GT-055 and GT-1 against carbapenem-resistant strains of *E. coli*, *K. pneumoniae*, and *A. baumannii*, including isolates that produce carbapenemases is unequivocal. Preclinical trials are being developed to determine the pharmacokinetic-pharmacodynamic properties of GT-055, and clinical assays were programmed to start in 2020 with the GT-1/GT-055 combination [67,169]. Table 4 reports the details of a preclinical trial that investigated the effects of GT-1 and GT-1/GT-055 in mouse infection models of *B. pseudomallei* and *Y. pestis*. Collectively, the efficacy of the combination was like that of ciprofloxacin in the *Y. pestis* model, while it was less effective in the *B. pseudomallei* infection model [113].

### 5.2. Boronic Acid Derivatives

Aiming at developing non-acylating BLEsIs, boron-based compounds were recently synthetized. Even if they were originally thought of as SBLEsIs, the most recent compounds have emerged also as MBLEsIs [170,171]. More useful than the acyclic boronates developed firstly, cyclic boronates proved to react very rapidly with BLEs forming stable enzyme-inhibitor complexes. Despite further studies being necessary to establish the kinetic mechanisms of the binding of bicyclic boronates to BLEs, results from microbiological assays already support the potential of these compounds as ESBLEsIs [172,173].

#### 5.2.1. Clinically Approved Boronic Acid Derivative

##### Vaborbactam (RPX-7009)

Vaborbactam (formerly RPX-7009) (C_12_H_16_BNO_5_S, MW = 297), IUPAC name 2-[(3R,6S)-2-hydroxy-3-[(2-thiophen-2-ylacetyl)amino]oxaborinan-6-yl]acetic acid, is the only boronate-based BLEsI approved for clinical application and already marketed. Structurally, it encompasses a six-term cyclic boronic acid pharmacophore functionalized with a residue of acetic acid on the carbon adjacent to the oxygen atom and with an amine group acylated with the 2-tyofen acetic acid on the carbon adjacent to the boron atom [114]. Vaborbactam has a high affinity for SBLEs, including carbapenemase produced by *K. pneumoniae*. Although developed to inhibit KPC-2 and KPC-3, vaborbactam proved to inhibit a variety of BLEs, including CTX-M-15, SHV-12, and TEMs of class A, as well as DHAs, MIRs, FOXs, and P99 of class C, which can give bacteria-resistance to broad-spectrum cephalosporins, with IC_50_ values in the nM range [67]. Concerning carbapenemases belonging to the family of SBLEs, boronic acid derivatives work by forming reversible covalent bonds with the hydroxyl group of the serine in their active site, mimicking the transition state that carbapenemases should form with BLAs to catalyze their hydrolysis [115]. Vaborbactam exhibited a good safety profile in human phase I clinical trials, and similar adverse events were observed in both placebo and treatment groups [116]. More details concerning this clinical trial are reported in Table 4 [116]. Varbobactam has been investigated in combination with meropenem in phase I clinical trials, registered at Clinical Trials.gov under identifier NCT02020434, to treat bacterial infections in subjects with varying degrees of renal insufficiency [114,117]. Table 4 reports details concerning the route of administration and pharmacokinetic data concerning the co-administration of a single dose of the combination meropenem/varbobactam to individuals without renal impairments [117].

Vaborbactam showed no antibacterial activity when administered alone. The molecule in combination with meropenem potentiated the bactericidal actions of meropenem against carbapenem-resistant KPC-producing *E. coli*, *K. pneumoniae*, and *E. cloacae* in a dose-dependent manner. Vaborbactam restored the antimicrobial activity of meropenem in animal models of infections caused by some meropenem non-susceptible KPC-producing *Enterobacteriaceae*. In August 2017, the combination vaborbactam/meropenem, currently marketed under the name Vabomere, was approved by the FDA for the treatment of adult patients with cUTIs [114]. Vabomere consists of vaborbactam and meropenem for intravenous administration. The efficacy and safety of meropenem/vaborbactam have been evaluated in two randomized clinical trials against antibiotic non-susceptible Gram-negative organisms of *E. coli*, *K. pneumoniae*, and *E. cloacae* named (TANGO) I (Clinicaltrial.gov identifier: NCT02166476) [118] and TANGO II (Clinicaltrial.gov identifier: NCT02168946) [119]. TANGO I demonstrated the efficacy of this new combination in cUTIs in comparison with piperacillin/tazobactam, whereas TANGO II demonstrated the efficacy of meropenem/vaborbactam in the treatment of cUTIs, HAP, VAP, cIAIs, or bloodstream infection caused by CRE. Moreover, the co-administration of varbobactam restored the MICs of meropenem in strains of *Enterobacteriaceae*, which showed decreased meropenem susceptibility due to the production of AmpCs or ESSBLEs [67]. However, vaborbactam was not able to improve the activity of meropenem against MDR non-fermenting Gram-negative isolates, such as *P. aeruginosa* and *Acinetobacter* spp. Also, varbobactam displayed weak potency against BLEs of class D, such as OXA-48, and was totally inactive against MBLEs of class B [67].

#### 5.2.2. Boronic Acid Derivatives in Experimental Phase

##### Taniborbactam (VNRX-5133)

Taniborbactam (formerly VNRX-5133), (C_19_H_28_BN_3_O_5_, MW = 389.3), IUPAC name (3R)-3-[[2-[4-(2-aminoethylamino)cyclohexyl]acetyl]amino]-2-hydroxy-3,4-dihydro-1,2-benzoxaborinine-8-carboxylic acid, is a BLEsI belonging to the cyclic boronate family, possessing broad-spectrum activity against KPCs, OXA-48 and MBLEs, including VIMs and NDMs, but not IMPs [120]. Structurally, it differs from varbobactam for the presence of a benzoic acid residue condensed on the boronate nucleus and of a 4-(2-aminoethylamino) cyclohexyl acetyl group acylating the amine residue [120]. Taniborbactam was the first inhibitor showing direct inhibitory activity against class-A, -B, -C, and -D enzymes. Particularly, taniborbactam inhibits both SBLEs and MBLEs, using different mechanisms. While it inhibits SBLEs with slow dissociation, it behaves as a reversible competitive inhibitor, with low inhibitor constant (K_i_) values and rapid dissociation towards MBLEs [121]. Taniborbactam in combination with cefepime and meropenem has been tested for the treatment of complicated infections caused by MDR pathogens such as CRE and CRPA, including strains expressing ESSBLEs, AmpCs, OXAs, KPCs, and MBLEs (VIMs NDMs) [67]. Particularly, cefepime–taniborbactam is a Venatorx intravenous combination that is being developed for the treatment of cUTIs, HAP, and VAP. Several phase I clinical trials have been undertaken to evaluate its pharmacokinetic and pharmacodynamic parameters and safety (registration No. NCT03870490, NCT03332732, and NCT02955459 at ClinicalTrials.gov) [120]. Details of the route of administration and pharmacokinetic parameters of clinical trial NCT02955459, which also evaluated the safety of taniborbactam/cefepime in healthy volunteers, are available in Table 4 [122]. The US FDA has granted QIDP and fast-track designations to cefepime–taniborbactam combination for the treatment of cUTIs, HAP, and VAP. Cefepime–taniborbactam is being considered in a phase III study called CERTAIN-1 (Cefepime Rescue with Taniborbactam in cUTI) (ClinicalTrials.gov NCT03840148), evaluating its safety and efficacy in adults with cUTIs, including APN and infections caused by MBLE-producing strains [120,123]. The trial is assessing the efficacy, safety, and tolerability of cefepime–taniborbactam compared to meropenem. Additionally, a phase III CERTAIN-2 study targeting HAP and VAP infections started in 2022. Notably, the FDA and the EMA approved Venatorx’s initial pediatric study plan (iPSP) and pediatric investigation plan (PIP) respectively, for cefepime–taniborbactam, thus enabling Venatorx and its partner, the Global Antibiotic Research and Development Partnership (GARDP), to initiate clinical trials for cefepime–taniborbactam in pediatric patients, including newborns [123]. In support of the PIP and iPSP, studies are underway to find the juvenile toxic dose for cefepime–taniborbactam. Taniborbactam is currently under investigation in clinical trial NCT03840148 (Safety and Efficacy Study of Cefepime/VNRX-5133 in Patients With Complicated Urinary Tract Infections).

##### Ledaborbactam Etzadroxil (Formerly VNRX-7145) as Orally Bioavailable Prodrug of Ledarbobactam (VNRX-5236)

VNRX-7145 (C_19_ H_26_ BNO_7_, MW = 391.2), IUPAC name 2-hydroxy-3-propionylamino-3,4-dihydro-2H-1-oxa-2-bora-naphthalene-8-carboxylic acid 2-ethyl-butyryloxymethyl ester, is a novel and not yet marketed cyclic boronate BLEsI with good oral bioavailability. Actually, VNRX-7145 is the prodrug of the real broad-spectrum boronic acid BLEsI (VNRX-5236), which is produced by in-vivo biotrasformation of VNRX-7145. Structurally, VNRX-7145 is like taniborbactam except for having a propionil acylating residue on the amine group and a 2-ethyl-butyryloxymethyl residue esterifying the carboxyl group on the phenyl ring. The in-vivo hydrolysis of the ester link by esterase enzymes sets free the active VNRX-5236 [124]. VNRX-5236 (C_12_ H_14_ B N O_5_, MW = 263.1), IUPAC name 2-hydroxy-3-propionylamino-3,4-dihydro-2H-1-oxa-2-bora-naphthalene-8-carboxylic inhibits ESSBLEs of classes A, C, and D, forming a covalent reversible bond between the boron atom and the hydroxyl group of the serine in the active site of the enzymes [67,125]. Specifically, the spectrum of inhibition of VNRX-5236 includes class C cephalosporinases, and classes A and D carbapenemases, such as KPC-2 and OXA-48, respectively (IC_50_ in the nM range) [126]. Currently, VNRX-7145 is being developed in combination with ceftibuten, a third-generation cephalosporin, due to its good oral bioavailability. The ceftibuten/VNRX-7145 combination is designed specifically to treat infections caused by MDR Gram-negative pathogens that are resistant to available oral and intravenous antibiotics, including fluoroquinolones, cephalosporins, and carbapenems [126]. Having proven modest activity against the enzymes of class B such as NDM-1 and VIM-2, the combination is particularly suitable to treat infections caused by *Enterobacteriaceae*, which produce carbapenemases and ESSBLEs [67]. Notably, both in-vitro and in-vivo studies established that VNRX-5236 is capable of restoring the activity of ceftibuten against such strains. The VNRX-7145 combination is now in Phase I studies to treat resistant UTIs. A phase I clinical trial to determine the safety and pharmacokinetic (PK) of VNRX-7145 administered as a single ascending dose (SAD) and multiple ascending doses (MAD) has been recently completed by Venatorx with positive top-line results, as announced in July 2021 (registration No. NCT04243863 at ClinicalTrials.gov) [126]. Particularly, no serious adverse event was observed, and VNRX-7145 was well-tolerated up to the highest single or multiple doses administered. Currently, a Phase I drug-drug interaction (DDI) study (ClinicalTrials.gov – NCT04877379), which will provide an initial assessment of the safety and PK of single and multiple doses of VNRX-7145 and ceftibuten, to treat resistant cUTIs is underway. Table 4 reports some details concerning the pharmacokinetic and oral availability of VNRX-7145 observed in a preclinical study in mice and rats [125].

##### Xeruborbactam (QPX-7728) 

Xeruborbactam (C_10_H_8_BFO_4_, MW = 221.98), IUPAC name 5-Fluoro-2-hydroxy-1,1a,2,7b-tetrahydro-3-oxa-2-bora-cyclopropa[a]naphthalene-4-carboxylic acid, is an ultra-broad-spectrum BLEsI discovered by Qpex scientists. Structurally, it consists of a benzocondensated cyclic boronic acid in turn condensed to a cyclopropane ring with a carboxylic group in position 4 and a fluorine atom in position 5 [127]. QPX-7728 inhibits key-note ESSBLEs and MBLEs including important carbapenemases, such as KPC-2, IMP-1, VIM-1, NDM-1, OXA-23, and OXA-48, at nano molar concentrations. As observed only for taniborbactam, this compound includes in its spectrum of inhibition also class-B and class-D enzymes and is less affected by porin modifications and efflux mechanisms [127]. Being both orally bioavailable and administrable via IVI, QPX-7728 is a promising agent for use in combination with several BLAs, including aztreonam, biapenem, meropenem, ceftibuten, ceftazidime, tebipenem, cefepime, ceftolozane, and ertapenem for the treatment of a wide range of MDR Gram-negative bacterial infections, by both intravenous and oral administration [127]. Particularly, the meropenem/QPX7728 combination proved to be much more potent than other combinations such as meropenem–vaborbactam, ceftazidime–avibactam, and imipenem–relebactam against all groups of MDR *Enterobacteriaceae* not susceptible to carbapenems including also those producing KPCs and MBLEs carbapenemases [67]. Promisingly, the meropenem/QPX-7728 combination was in-vitro effective against several representatives of carbapenem-resistant strains of *A. baumannii* producing CHDL-, NDM-, and KPC-like carbapenemases and KPC-producing isolates of *P. aeruginosa* [128,129]. 

In preclinical assays, QPX7728 has demonstrated a profile that exceeds that of recently marketed BLEsIs, as well as that of those in ongoing clinical trials [130]. Table 4 reports some pharmacokinetic data concerning QPX-7728 intravenous administration in combination with several BLAs [131]. Particularly, QPX-7728 restored and increased the efficacy of several BLAs against KPC-producing *Enterobacteriaceae* in a neutropenic mouse thigh infection model and against *Acinetobacter* and *Pseudomonal* infections [130,174]. On 13 May 2021, for QPX-7728, Qpex Biopharma began enrolling in the phase I trial for bacterial infections (in volunteers) in the USA (PO). On 17 May 2021, Qpex Biopharma planned a phase I trial in the USA in the second half of 2021, and on 19 Jan 2022, xeruborbactam received QIDP status for bacterial infections in the USA [174].

Qpex Biopharma is currently conducting a Phase I study that is investigating the safety and pharmacokinetics of QPX7728 alone in healthy adult subjects, following single and multiple intravenous doses and in combination with a BLA [130].

QPX-7728 currently enables two product candidates in the Qpex pipeline. OMNIvance™, which is an IV-administered QPX-7728-based product with best-in-class coverage of key pathogens, including carbapenem-resistant *Acinetobacter*, *Enterobacteriaceae*, and *Pseudomonas* spp. [130].

ORAvance™, which instead is an orally-administered combination product based on QPX-7728 to treat infections that occur in the outpatient and community setting caused by MDR Gram-negative bacteria, including *Enterobacteriaceae* producing ESSBLEs and carbapenemases [130].

### 5.3. Thiazole-Carboxylates Derivative

#### ANT-2681 

ANT-2681 (C_12_H_12_F_2_N_8_O_4_S_2_, MW = 434.4) is a small molecule being developed by Antabio as a novel BLEsI. Structurally, it is very different from the other novel inhibitors previously described. ANT-2681 is a thiazole carboxylate derivative consisting of a 3,5-difluoro benzenesulfonamide nucleus linked to a 4-thiazol-carbamide in position 5 of the heterocycle with an ureidoguanidine residue in position 4 [132]. The structure of ANT-2681 represents the conclusion of a research study in medicinal chemistry, which had previously led to the detection of an advanced lead compound containing pyridine, ANT-431 (Figure 6), which demonstrated efficacy in a mouse infection model, by NDM-positive carbapenem-resistant isolates of *Enterobacteriaceae* (CRE) [132]. 

Promisingly, ANT-2681 inhibited specifically the enzymatic activity of MBLEs by interacting with the dinuclear zinc ion cluster present in the active site of these enzymes. Preclinical studies are underway to assess its efficacy in treating complicated infections by Gram-negative bacteria producing MBLEs [133]. In this regard, on 21 Jun 2019 ANT-2681 received QIDP status for UTIs (Combination therapy, Treatment-resistant) in the USA, while on 18 Apr 2020, antimicrobial, as well as pharmacodynamic and pharmacokinetic data from preclinical studies in Gram-negative infection were presented at the 30th European Congress of Clinical Microbiology and Infectious Diseases (ECCMID-2020) [133]. 

In current preclinical studies, ANT-2681 was associated with meropenem. Particularly, the pharmacokinetics/pharmacodynamics of meropenem/ANT-2681 (MEM/ANT-2681) were studied in a murine neutropenic thigh model for the treatment of infections caused by NDM-producing *Enterobacteriaceae* [134]. Dose-ranging studies were performed with both meropenem and ANT2681. Half-maximal effect was observed with ANT-2681 at a dose of 89 mg/kg every 4h intravenous infusion [134].

Moreover, preclinical susceptibility studies have been performed with MEM/ANT-2681 against several MBLE-positive *Enterobacteriaceae*, including many NDM-CRE. Notably, the addition of ANT-2681 at 8 μg/mL of MEM reduced the MEM MICs_90_ from >32 μg/mL to 8 μg/mL. Furthermore, by associating MEM/ANT-2681 1/1 (8 μg/mL) the *Enterobacteriaceae* positive for the Verona integron-encoded MBLE (VIM) were reduced by 74.9%, while those positive for the imipenem hydrolyzing MBLE (IMP) were decreased by 85.7%.

Very interestingly, MICs much lower than those of other combinations such as aztreonam/avbactam, cefiderocol/taniborbactam, and cefepime/taniborbactam were observed against NDM-positive isolates of *E. coli* (MIC_90_ = 1 μg/mL vs. 4 μg/mL, >32 μg/mL, and >32 μg/mL, respectively) [135].

## 6. Works in Progress

With the aim of inhibiting carbapanemases, other β-lactam-derived inhibitors are in early stages of development. Within the group of C-6 substituted penicillin sulfones, it has been observed that a certain specificity of inhibition can be obtained by changing substituents and good activity against some class B BLEs was observed [159,175]. Among C-2 substituted carbapenem compounds J-110,441 and J-111,225 which are 1β-methylcarbapenems with a benzothenyl moiety and a trans-3,5 disubstituted pyrrolidinylthio moiety in C-2, respectively inhibited, among others, CcrA and IMP-1 MBLEs and showed intrinsic bactericidal activity against *S. aureus* and *P. aeruginosa* [176,177,178]. Among non-β-lactam-derived inhibitors, bisthiazolidines (namely L-CS319, D-CS319, L-VC26, and D-VC26) showed high affinity for the Zn(II) binding group, due to the presence of a free thiol group thus succeeding in inhibiting MBLEs [179]. Dipicolinic acid derivatives and triazole-containing molecules are other interesting potential inhibitors of MBLEs, whose efficacy remains to be demonstrated in preclinical trials [180,181]. Particularly interested readers can find more details about these and other new inhibitory compounds still in development in several relevant papers [182,183,184,185,186,187,188].

### Evolving Resistance to Recently Approved BLAs/BLEsIs Combinations

Clinically approved combinations such as ceftazidime/avibactam, imipenem/relebactam, and meropenem/vaborbactam provided effective therapeutic options to solve difficult-to-treat infections by clinically relevant superbugs such as carbapenemase-producing or carbapenem-resistant Gram-negative pathogens. Ceftazidime/avibactam combination has the broadest spectrum of activity, being effective against pathogens resistant to carbapenem and producing KPCs, OXA-48-like ESSBLEs, and AmpCs. Similarly, the association of relebactam to imipenem led to improved activity against *Enterobacteriaceae* producing KPCs and carbapenem-resistant *P. aeruginosa* but did not improve imipenem activity against isolates that produce OXA-48-like enzymes. Meropenem/vaborbactam, in addition to being ineffective on isolates producing OXA-48, also failed against *P. aeruginosa* isolates, which are meropenem-resistant regardless of their capacity to produce BLEs [189]. 

These new approved combinations go on to be ineffective on MBLEs, against which no approved inhibitor works and can fail against strains initially susceptible to developing resistance during therapy. From the year of its approval (2015) to date, by various molecular mechanisms, several isolates of *Enterobacteriaceae* and particularly of *K. pneumoniae*, as well as of *P. aeruginosa* have developed resistance to the ceftazidime/avibactam combination [190,191]. The most-reported processes leading to isolates resistant to ceftazidime/avibactam regarded modifications of either KPC-2 and KPC-3 BLEs, the structural modification of CTX-M-like or AmpC enzymes [67], and the increased KPC expression associated with the reduction of porins and the enhancement of AcrAB-TolC efflux pump [67]. Similarly, the development of resistance to ceftazidime/avibactam during the therapies adopted to treat infections by *P. aeruginosa* was associated with variants of extended-spectrum OXA-2 or OXA-10 such as OXA-539, OXA-681, and OXA-14, which added to variants of AmpCs determined resistance also to the recently developed ceftolozane/tazobactam combination [192]. Concerning carbapenems/BLEsIs combinations, the emergence of both meropenem/vaborbactam and imipenem/relebactam resistant isolates of *Enterobacteriaceae* and *K. pneumoniae* has been associated with mutations in porins often combined with an increased blaKPC-copy number and enhanced efflux via the efflux pump AcrAB-TolC efflux [193,194]. In this scenario, judicious use of these agents and continuous surveillance in future years is encouraged to prevent the emergence and spread of resistance to these new combinations among target pathogens.

## 7. Summary and Authors’ Conclusions 

As the present work highlighted, the sector of BLAs, BLEs, and of BLEsIs is very vast. 

At the basis of this vastness and complexity lies mainly the fact that both the number of BLE-producing bacteria and that of BLEs themselves are constantly evolving. On one hand, the genes encoding for BLEs are easily transmittable between bacteria, especially if located in plasmids, thus improving continuously the number of pathogens enabled to produce BLEs. On the other hand, these enzymes commute continuously giving rise to an expanding number of new variants with a progressively broader spectrum of action, thus becoming capable of inactivating even the most powerful and latest generation antibiotics/inhibitors combinations.

From the information reported in this review, it emerges that the spread of the genes encoding for BLEs, as well as the genetic mutations leading to new enzymes not susceptible to the inhibitors currently available, occur much more quickly than the discovery and approval of new BLEsIs, with adequate spectra of action. The state of the art highlights the existence of an unequal struggle between a small number of clinically approved inhibitors and an immense and increasing number of different types of BLEs. In this regard, the graph bar in Figure 7 points out that only a very low number of BLEsIs developed so far are already clinically approved and are available for use in therapy (27.3%), while most of them are still in the preclinical (27.3%) or clinical trials (36.2%), or even at very early stages of development (9%).

Moreover, considering that the major concern consists of the increasing emergence of bacteria resistant to carbapenems, mainly due to the production of carbapenemases, such as VIMs, IMPs, NDMs, KPCs, and OXAs, the number of functioning inhibitors clinically applicable reduces dramatically.

As can be seen in Figure 8, among the six products currently on the market (27.3%, Figure 7), no compound is active on MBLEs of class B, including NDM-1, able to hydrolyze all available antibiotics except for aztreonam. Only three compounds (50%) inhibit class-A carbapenemases (KPCs), and among these, only avibactam is active also on some OXA-like carbapenemases of class D, thus establishing that 50% of clinically applied inhibitors are unable to restore the activity of carbapenems.

The scenario improves if the molecules not yet approved are also considered. As shown in Figure 9, although most of the developed BLEsIs/BLAs combinations possess activity only on ESSBLEs and not on carbapanemases of MBLEs, good percentages of the developed associations were found to be active on the most relevant OXA-like carbapenemases of class D (56.4%) and on the class-A carbapenemases of KPC family (74.5%). Among all the available approved and not yet approved inhibitors, the long-established and clinically approved BLEsIs, such as CA, sulbactam, and tazobactam are ineffective on carbapenemases, which are inhibited by the more recently developed BLEsIs, among which only three compounds have been recently approved and are currently applied in therapy (avibactam, relebactam, and vaborbactam). Concerning the 12 more promising carbapanemase inhibitors developed so far, 9 out of 12 compounds are in clinical or preclinical trials. The percentages of molecules active on the most clinically significant carbapenemases are observable in Figure 10.

Almost all the new BLEsIs are capable of inhibiting KPCs, more than 50% of inhibitors work well against OXA-48, while the percentages of compounds are active against other class-D carbapenemase and above all against carbapenemases of class-B dramatically decrease.

When the available combinations BLAs/BLEsIs, developed using the 12 inhibitors previously considered, were tested against carbapenem-resistant *A. baumannii* (CRAB), *P. aeruginosa* (CRPA), and *Enterobacteriaceae* (CRE) producing either SBLEs or MBLEs, CRE were the most susceptible bacteria while CRPA and CRAB were less susceptible. The percentages of combinations effective on bacteria producing SBLEs were higher than those effective against MBLE-producers. Particularly, Figure 11a,b reports the percentages of combinations active on the different bacterial species producing SBLEs (Figure 11a) and those of combinations that are affective on the different bacterial species producing MBLEs (Figure 11b).

Unfortunately, among the clinically approved BLEsIs/BLAs, none were active on carbapenem-resistant bacteria producing MBLEs, and against *A. baumannii* producing SBLEs, while only one was active against CRPA and CRE producing SBLEs.

The continuous search for new inhibitors must therefore be strongly encouraged and supported as it represents the main and fundamental challenge for future successful therapeutic regimens.

## Data Availability

Not applicable.

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
