# Peer review of "β-Lactam Antibiotics and β-Lactamase Enzymes Inhibitors, Part 2: Our Limited Resources"

_pharmaceuticals, 2022, doi:10.3390/ph15040476_

Round 1

Reviewer 1 Report

In this manuscript, authors have worked on β-Lactam Antibiotics and β-Lactamase Enzyme Inhibitors, Part 2: Our Limited Resources. Following are some points needs to be considered.

  1. Abstract Line 22 NMD-1 what does it abbreviates for?
  2. Abstract line 14 to 16 is not clear.
  3. Abstract line 17-18 grammatical error.
  4. Introduction line 42-45. References are required.
  5. Structures in figure 1 and 2 could be drawn through some software tool.
  6. Most part of introduction has only ref 1 repeated. Adding more recent references will be beneficial.
  7. Line 78 “Anyway…. “Can be removed.
  8. Introduction is lengthy and to explain the BLE mechanism a figure could be incorporated which would aid the understanding.
  9. Table 1 and 2 and 3 needs to be adjusted to fit to window as information is not shown in full.
  10. Scheme 1 needs to be had a reference if it has been taken from some article.
  11. Referencing needs to be checked again as certain references are repeated again and again like ref no 3, 1, 66,
  12. Headings “Not Clinically Approved Boronic Acid Derivatives” can be reframed to appear more apt to the topic. The negative sentences could be reframed to be assertive one.
  13. Line spacing of the entire manuscript needs to be checked as it is not uniform.
  14. Figure 9 and 10 needs to be resolved to remove the blurr.
  15. Reference section needs to be reformatted to remove unwanted elements like there is line appearing in between ref 3 and 4, 28 and 29, 57 and 58.
  16. Ref 59, 61, 64 are of Drug Banck online only. Although for different drugs so can be merged at one place.
  17. Ref 98, 99, 103, 105 are same sites although accessed for different chemicals so can be merged.

Author Response

In this manuscript, authors have worked on β-Lactam Antibiotics and β-Lactamase Enzyme Inhibitors, Part 2: Our Limited Resources. Following are some points needs to be considered.

Abstract Line 22 NMD-1 what does it abbreviates for?

We apologise with the Reviewer, “NMD-1” was incorrect. The correct abbreviation is NDM-1, whose meaning is specified in the previous lines. The error has been corrected.

Abstract line 14 to 16 is not clear.

The sentence has been reformulated for more clarity.

Abstract line 17-18 grammatical error.

The Reviewer is right. The error has been corrected.

Introduction line 42-45. References are required.

The required reference (new Ref. 3) has been included.

Structures in figure 1 and 2 could be drawn through some software tool.

Indeed, the chemical structures in Figure 1 and Figure 2 have been drawn using the ChemDraw Ultra 7.0 software (Chem Office 2002). Such information has been inserted in the captions of Figure 1 and 2 and in all the captions of Figures or Schemes where chemical structures are present.

Most part of introduction has only ref 1 repeated. Adding more recent references will be beneficial.

As requested, the citation of Ref. 1 was limited to the first few sentences (four times).

Line 78 “Anyway…. “Can be removed.

As requested, “Anyway” has been removed.

Introduction is lengthy and to explain the BLE mechanism a figure could be incorporated which would aid the understanding.

We make kindly note to the Reviewer that the mechanism of BLEs has already been graphically explained in Section 2 dedicated to BLEs (Scheme1 and Scheme 2), both for SBLEs and MBLEs.

Table 1 and 2 and 3 needs to be adjusted to fit to window as information is not shown in full.

The signalled Tables have been adjusted. We checked the pdf file and now all the information contained in Tables is clearly visible.

Scheme 1 needs to be had a reference if it has been taken from some article.

Both Scheme 1 and Scheme 2 have been realized by us using the ChemDraw Ultra 7.0 software (Chem Office 2002). Anyway, this detail has been specified in the caption of each Scheme.

Referencing needs to be checked again as certain references are repeated again and again like ref no 3, 1, 66,

When possible, we have reduced excessive repetitions of Refs. But since the review contains already 194 references, we have preferred not to add further ones.

Headings “Not Clinically Approved Boronic Acid Derivatives” can be reframed to appear more apt to the topic. The negative sentences could be reframed to be assertive one.

As requested, all the negative sentences have been reframed to be assertive.

Line spacing of the entire manuscript needs to be checked as it is not uniform.

Line spacing has been uniformed.

Figure 9 and 10 needs to be resolved to remove the blurr.

As requested, the signalled Figures have been remade. The quality of Figure 9 and Figure 10 has been improved and blur has been removed.

Reference section needs to be reformatted to remove unwanted elements like there is line appearing in between ref 3 and 4, 28 and 29, 57 and 58.

The reference section has been checked and reformatted. The unwanted elements have been removed.

Ref 59, 61, 64 are of Drug Banck online only. Although for different drugs so can be merged at one place.

These Refs. are all of Drug Bank, but the links are different. So, they cannot be merged.

Ref 98, 99, 103, 105 are same sites although accessed for different chemicals so can be merged.

These Refs. contain different links to the web sites. So, they cannot be merged.

Reviewer 2 Report

paper by Alfei and Schito presents a long story entitled "β-Lactam Antibiotics and β-Lactamase Enzyme Inhibitors, Part 2: Our Limited Resources". β-lactam antibiotics (BLAs) are crucial molecules among antibacterial drugs, but the increasing emergence of resistance to them, developed by bacteria producing β-lactamase enzymes (BLEs), is becoming one of the major warnings to global public health. this work shows a lot of aspects regarding β-Lactam Antibiotics and β-Lactamase Enzyme Inhibitors. in my opinion, the work, does not have flaws but is too long - and some parts of it could be shortened. Additionally, it will be necessary to correct the grammar of the work

Author Response

paper by Alfei and Schito presents a long story entitled "β-Lactam Antibiotics and β-Lactamase Enzyme Inhibitors, Part 2: Our Limited Resources". β-lactam antibiotics (BLAs) are crucial molecules among antibacterial drugs, but the increasing emergence of resistance to them, developed by bacteria producing β-lactamase enzymes (BLEs), is becoming one of the major warnings to global public health. this work shows a lot of aspects regarding β-Lactam Antibiotics and β-Lactamase Enzyme Inhibitors. in my opinion, the work, does not have flaws but is too long - and some parts of it could be shortened. Additionally, it will be necessary to correct the grammar of the work

We thank a lot the Reviewer for having appreciated our work. We have checked our manuscript carefully, establishing that all information contained is essential for its completeness. In addition, the manuscript has been revised by the native English teacher Deirdre Kantz, who works for both University of Genoa and Pavia, which helped us in correcting the grammar.